# Boosting Multi-Domain Reasoning of LLMs via Curvature-Guided Policy Optimization

**Xize Liang**[1][*][†], **Lin Yang**[2][†], **Jie Wang**[1][‡], **Rui Liu**[1], **Yang Lu**[1], **Jinliang Zeng**[1],
**Hanzhu Chen**[1], **Dong Li**[2], **Jianye Hao**[3]
[1]MoE Key Laboratory of Brain-inspired Intelligent Perception and Cognition,
University of Science and Technology of China
[2]Huawei Technologies Co., Ltd.
[3]College of Intelligence and Computing, Tianjin University
`xizeliang@mail.ustc.edu.cn`

## Abstract

Multi-domain reinforcement learning (RL) for large language models (LLMs) involves highly intricate reward surfaces, posing significant challenges in finding parameters that excel across all domains. Recent empirical studies have further highlighted conflicts among domains, where gains in one capability often come at the expense of another. However, approaches to mitigate such conflicts and enhance multi-domain reasoning remain largely underexplored. To address this challenge, we propose **C**urvature-**G**uided **P**olicy **O**ptimization (**CGPO**), a principled and scalable training framework to advance the multi-domain reasoning of LLMs. Inspired by Newton's method, CGPO exploits the geometric structure in the reward surface, while sidestepping the prohibitive cost of Hessian computation. At each update, CGPO processes domains in random order, preconditioning their gradients with curvature information from other domains to foster richer cross-domain interactions. This mechanism further promotes implicit gradient alignment by maximizing inter-domain inner products in expectation, steering the parameters toward regions that jointly enhance multi-domain performance. Extensive experiments on a mixed dataset covering math, coding, science, and creative writing, evaluated across seven widely-used benchmarks, show that CGPO significantly outperforms all baselines in terms of faster reward improvement and stronger multi-domain capability.

## 1 Introduction

Large language models (LLMs) have recently achieved remarkable progress in complex reasoning, including mathematical problem solving (Yang et al., 2024; Yu et al., 2025a), code generation (Ye et al., 2025; Zeng et al., 2025), circuit generation (Bai et al., 2026), optimization modeling (Liu et al., 2026), and creative writing (Fein et al., 2025; Carrera et al., 2025). A key driver behind these advances is reinforcement learning (RL), particularly policy optimization methods such as PPO (Schulman et al., 2017) and GRPO (Shao et al., 2024). While earlier work primarily focused on applying RL within single domains (Hu et al., 2025; Yu et al., 2025a), more recent studies have moved toward multi-domain reasoning, constructing diverse datasets (Cheng et al., 2025), training general reward models (Ma et al., 2025), and empirically examining interactions among different reasoning capabilities (Li et al., 2025b; Cheng et al., 2025).

Despite these advances, multi-domain RL for LLMs still confronts significant challenges. The coexistence of diverse data distributions and reward signals produces highly complex reward surfaces, making it difficult to find parameters that excel across all domains simultaneously (Vithayathil Varghese & Mahmoud, 2020; Crawshaw, 2020). Recent studies further show that, although multi-domain RL can yield overall benefits, it is often hindered by cross-domain conflicts, where gains in one capability are accompanied by losses in another (Cheng et al., 2025; Li et al., 2025b). These difficulties

---

[*]Work conducted during the research internship of Xize Liang (xizeliang@mail.ustc.edu.cn) at Huawei.
[†]Equal contribution.
[‡]Corresponding author. Email: jiewangx@ustc.edu.cn.

are further compounded by the nature of RL training: on one hand, online sampling (i.e., rollouts) introduces unpredictable interactions among domain-specific samples; on the other hand, generating rollouts is computationally expensive, and much of this effort is wasted when cross-domain conflicts cancel out the contributions. These considerations make it crucial to develop RL frameworks that fully exploit mixed datasets to enhance LLMs' reasoning across diverse domains.

Cross-domain conflicts often manifest as gradient conflicts (Chen et al., 2025b; Liang et al., 2025), yet widely-used approaches for mitigating them face notable limitations in the context of RL for LLMs. Most existing methods intervene during gradient aggregation once conflicts occur, aiming to balance updates across domains. On the one hand, they do not leverage the underlying geometry of the reward surface or loss landscape (Liu et al., 2023; Sener & Koltun, 2018). On noisy, rollout-based gradients, such purely reactive strategies tend to amplify update variance and degrade both stability and performance. On the other hand, many techniques require storing and manipulating all domain gradients simultaneously on the device (Yu et al., 2020; Liu et al., 2024; 2021). This incurs substantial memory overhead that grows rapidly with the number of domains and can even result in out-of-memory failures, severely limiting the scalability of multi-domain RL for LLMs. Alternatively, recent work suggests that second-order methods such as Newton's method and its approximation SOAP (Vyas et al., 2025) can mitigate gradient conflicts in PINNs (Wang et al., 2025), but their reliance on Hessian computations renders them infeasible for the high-dimensional, rollout-heavy setting of RL for LLMs. These limitations compellingly motivate the following question: ***How to mitigate cross-domain conflicts in a manner that is both consistent with the nature of RL and efficient at scale, thereby enhancing the multi-domain reasoning capabilities of LLMs?***

In this paper, we propose CGPO, a principled and scalable policy optimization framework, to enhance multi-domain reasoning for LLMs[1]. CGPO draws inspiration from Newton's method, while incorporating a design specifically adapted to the distinct challenges of multi-domain RL for LLMs. Newton's method exploits the geometric structure of the loss landscape (i.e., the Hessian matrix) to precondition gradients, correcting directional deviations induced by anisotropy and facilitating efficient convergence. To retain these benefits while circumventing the computational burden of full Hessian computation, we adapt the preconditioning step into a lightweight mechanism tailored for efficient RL training of LLMs. Specifically, at each parameter update, domains are processed in random order, with each domain's gradient modulated by curvature information from others, thereby inducing rich cross-domain interactions. Another appealing feature of this mechanism is that it implicitly aligns domain gradients by maximizing their inner products in expectation, guiding the parameters toward regions of high cross-domain consistency. We validate CGPO on a diverse dataset of 20k samples spanning **mathematical reasoning, code generation, scientific QA, and creative writing** using Qwen2.5-3B-Instruct and Qwen2.5-7B-Instruct, evaluated across **seven benchmarks**. Our results demonstrate that CGPO consistently outperforms a broad spectrum of baselines—including curriculum learning strategies, gradient balancing techniques, and joint learning—achieving faster reward gains and markedly stronger multi-domain reasoning capabilities.

## 2 PRELIMINARIES

### 2.1 MULTI-DOMAIN LANGUAGE MODELING AS REINFORCEMENT LEARNING

An LLM $\pi_\theta$ (with parameters $\theta$) defines a conditional probability distribution over output responses $\mathbf{y} = [y_1, \ldots, y_T]$ given a query $\mathbf{x} \sim \mathcal{D}$, represented as $\pi_\theta(\mathbf{y} \mid \mathbf{x}) = \prod_{t=1}^{T} \pi_\theta(y_t \mid \mathbf{x}, \mathbf{y}_{1:t-1})$. To align LLMs with desired behaviors, recent work formulates language generation as a reinforcement learning (RL) problem. The model acts as a policy that interacts with an environment by generating responses $\mathbf{y}$ to queries $\mathbf{x}$, and each response receives a reward $R(\mathbf{x}, \mathbf{y}) \in \mathbb{R}$ that reflects its quality.

In many real-world applications, LLMs are expected to perform well across multiple domains, each corresponding to a distinct type of query or task. Formally, let there be $K$ domains with query distributions $\{\mathcal{D}_k\}_{k=1}^{K}$. Each domain $k$ defines its own reward function $R_k(\cdot, \cdot)$, reflecting task-specific quality criteria. Assuming equal importance for all domains, the multi-domain training objective is to maximize the average expected reward (we abbreviate $\mathbf{y} \sim \pi_\theta(\cdot \mid \mathbf{x})$ as $\mathbf{y} \sim \pi_\theta$): $\mathcal{J}(\theta) = \frac{1}{K} \sum_{k=1}^{K} \mathcal{J}_k(\theta) = \frac{1}{K} \sum_{k=1}^{K} \mathbb{E}_{\mathbf{x} \sim \mathcal{D}_k, \mathbf{y} \sim \pi_\theta}[R_k(\mathbf{x}, \mathbf{y})]$. We provide a discussion on extending this formulation to non-uniform domain importance in Appendix D.2.

---

[1]Additional discussion on the applicability of CGPO to LLM pre-training is provided in Appendix D.1.

## 2.2 POLICY OPTIMIZATION ALGORITHMS

The multi-domain formulation in Section 2.1 reduces to the standard RL objective when expressed with a generic query distribution $\mathcal{D}$ and reward function $R$, i.e., $\mathcal{J}(\theta) = \mathbb{E}_{\mathbf{x}\sim\mathcal{D},\mathbf{y}\sim\pi_\theta}[R(\mathbf{x},\mathbf{y})]$.

Directly optimizing $\mathcal{J}(\theta)$ is challenging due to the discrete, variable-length output space and the dependency of the distribution $\pi_\theta$ on the parameters $\theta$. Instead, the *policy gradient theorem* (Sutton et al., 1998) provides an unbiased estimator for the gradient, i.e., $\nabla_\theta \mathcal{J}(\theta) = \mathbb{E}_{\mathbf{x}\sim\mathcal{D},\mathbf{y}\sim\pi_\theta}[\nabla_\theta \log \pi_\theta(\mathbf{y}\mid\mathbf{x})A(\mathbf{x},\mathbf{y})]$, where $A(\mathbf{x},\mathbf{y}) = R(\mathbf{x},\mathbf{y}) - b(\mathbf{x})$ denotes the advantage of response $\mathbf{y}$ over a baseline $b(\mathbf{x})$. In practice, the true advantage function is unknown and must be estimated from rollouts. This is typically done by training a value function $V_\phi(\mathbf{x})$ to approximate the expected reward, and then computing an *estimated advantage* $\hat{A}(\mathbf{x},\mathbf{y}) = R(\mathbf{x},\mathbf{y}) - V_\phi(\mathbf{x})$. By combining this estimator with importance sampling using rollouts from an old policy $\pi_{\theta_{\text{old}}}$, one can define a *surrogate objective* $L(\theta;\theta_{\text{old}},\mathcal{D}) = \mathbb{E}_{\mathbf{x}\sim\mathcal{D},\mathbf{y}\sim\pi_{\theta_{\text{old}}}}\left[\frac{\pi_\theta(\mathbf{y}|\mathbf{x})}{\pi_{\theta_{\text{old}}}(\mathbf{y}|\mathbf{x})}\hat{A}(\mathbf{x},\mathbf{y})\right]$.

While the theoretical surrogate objective using the true advantage $A$ has a gradient that coincides exactly with $\nabla_\theta \mathcal{J}(\theta)$ at $\theta = \theta_{\text{old}}$, practical objectives using the estimated advantage $\hat{A}$ serve as a first-order approximation. This approximation is reliable as long as the updated policy $\pi_\theta$ remains close to $\pi_{\theta_{\text{old}}}$. Building on this, Proximal Policy Optimization (PPO) (Schulman et al., 2017) ensures stable policy updates by maximizing a clipped surrogate objective $L_{\text{PPO}}(\theta;\theta_{\text{old}},\mathcal{D}) = \mathbb{E}_{\mathbf{x}\sim\mathcal{D},\mathbf{y}\sim\pi_{\theta_{\text{old}}}}\left[\min\left(\frac{\pi_\theta(\mathbf{y}|\mathbf{x})}{\pi_{\theta_{\text{old}}}(\mathbf{y}|\mathbf{x})}\hat{A}(\mathbf{x},\mathbf{y}), \text{clip}_{1-\varepsilon}^{1+\varepsilon}\left(\frac{\pi_\theta(\mathbf{y}|\mathbf{x})}{\pi_{\theta_{\text{old}}}(\mathbf{y}|\mathbf{x})}\right)\hat{A}(\mathbf{x},\mathbf{y})\right)\right]$, where $\varepsilon$ is a small hyperparameter and $\text{clip}_{\gamma_{\text{low}}}^{\gamma_{\text{high}}}(\cdot) = \text{clip}(\cdot,\gamma_{\text{low}},\gamma_{\text{high}})$ is the clipping function.

However, the reliance of PPO on a separately trained critic model to estimate $b(\mathbf{x})$ introduces substantial memory and computational overhead. To address this, recent critic-free methods represented by GRPO (Shao et al., 2024) have emerged. GRPO estimates the baseline directly from a group of sampled responses. Specifically, it samples $G$ responses $\{\mathbf{y}^{(i)}\}_{i=1}^G$ for each query $\mathbf{x}$, obtains their rewards $\{r^{(i)}\}_{i=1}^G$, and then computes a normalized advantage for each response: $\hat{A}^{(i)} = \left[r^{(i)} - \text{mean}\left(\{r^{(j)}\}_{j=1}^G\right)\right]/\text{std}\left(\{r^{(j)}\}_{j=1}^G\right)$. The overall GRPO surrogate objective is

$$L_{\text{GRPO}}(\theta;\theta_{\text{old}},\mathcal{D}) = \mathbb{E}_{\mathbf{x}\sim\mathcal{D},\{\mathbf{y}^{(i)}\}_{i=1}^G\sim\pi_{\theta_{\text{old}}}}$$

$$\left[\frac{1}{G}\sum_{i=1}^G \min\left(\frac{\pi_\theta(\mathbf{y}^{(i)}\mid\mathbf{x})}{\pi_{\theta_{\text{old}}}(\mathbf{y}^{(i)}\mid\mathbf{x})}\hat{A}^{(i)}, \text{clip}_{1-\varepsilon_{\text{low}}}^{1+\varepsilon_{\text{high}}}\left(\frac{\pi_\theta(\mathbf{y}^{(i)}\mid\mathbf{x})}{\pi_{\theta_{\text{old}}}(\mathbf{y}^{(i)}\mid\mathbf{x})}\right)\hat{A}^{(i)}\right) - \beta\mathbb{D}_{\text{KL}}^{(i)}(\pi_\theta\|\pi_{\text{ref}})\right], \quad (1)$$

where $\varepsilon_{\text{low}}$, $\varepsilon_{\text{high}}$, and $\beta$ are hyperparameters, $\pi_{\text{ref}}$ is a reference policy (typically the initial model), and $\mathbb{D}_{\text{KL}}^{(i)}(\pi_\theta\|\pi_{\text{ref}})$ is a sample-based KL divergence penalty. In this work, we adopt GRPO as our base policy gradient algorithm due to its efficiency and scalability.

**Surrogate Objectives as Faithful Gradient Approximators.** While the policy gradient theorem provides an unbiased gradient for the true advantage $A$, practical algorithms rely on estimated advantages $\hat{A}$, which introduce variance. Surrogate objectives like PPO and GRPO are designed to stabilize these gradients: PPO uses clipping to enforce a trust region, making $\nabla_\theta L_{\text{PPO}}(\theta;\theta_{\text{old}},\mathcal{D})$ a reliable approximation of $\nabla_\theta \mathcal{J}(\theta)$, while GRPO's combination of clipping and KL regularization similarly produces a stable gradient $\nabla_\theta L_{\text{GRPO}}(\theta;\theta_{\text{old}},\mathcal{D})$ that approximates the KL-regularized objective $\nabla_\theta(\mathcal{J}(\theta) - \beta'\mathbb{D}_{\text{KL}}(\pi_\theta\|\pi_{\text{ref}}))$.

## 2.3 NEWTON'S METHOD FOR GRADIENT PRECONDITIONING

Newton's method is a classical second-order optimization algorithm that leverages the curvature of the objective to accelerate convergence. Given a twice-differentiable loss $L(\theta)$, the Newton update is $\theta_{t+1} = \theta_t - \mathbf{H}(\theta_t)^{-1}\mathbf{g}(\theta_t)$, where $\mathbf{g}(\theta_t) = \nabla_\theta L(\theta_t)$ and $\mathbf{H}(\theta_t) = \nabla_\theta^2 L(\theta_t)$ is the Hessian. By preconditioning the gradient with local curvature, Newton's method corrects for anisotropy, producing more direct steps toward an optimum. It is particularly effective in complex, conflicting landscapes; e.g., Wang et al. (2025) shows that Newton's method and its approximate variant SOAP (Vyas et al., 2025) mitigate gradient conflicts in PINNs and accelerate convergence.

However, directly applying Newton's method to RL for LLMs is impractical: the Hessian is high-dimensional and costly to compute or invert, and rollout-based gradients are noisy. Still, the principle

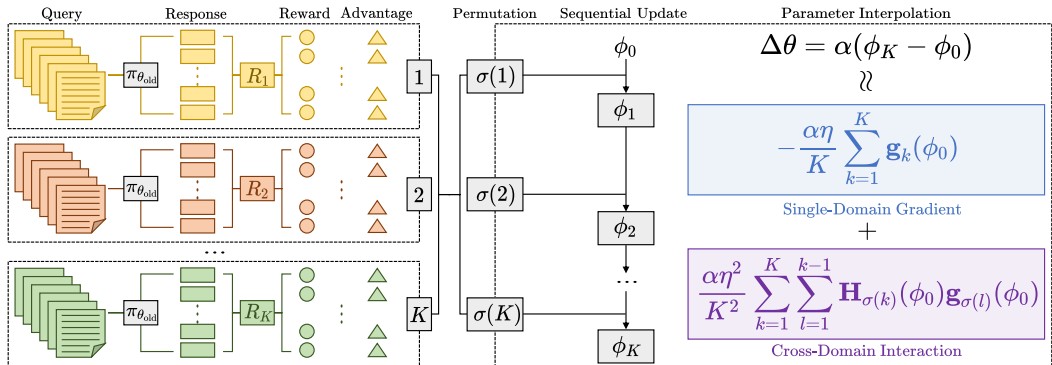

Figure 1: Illustration of CGPO (one update step). After generating responses, computing rewards, and estimating advantages for each domain, CGPO randomly permutes the domain order and applies updates sequentially, followed by interpolation with the original model. The parameter change $\Delta\theta$ can be approximately decomposed into a single-domain gradient term—capturing per-domain learning—and a cross-domain interaction term that facilitates transfer across domains. **Note that CGPO introduces only negligible additional computation overhead** (see Section 4.3 for details).

of leveraging curvature to guide updates provides a valuable foundation for designing optimization strategies that handle conflicting gradients and complex surfaces, as we explore in Section 3.

## 3 CURVATURE-GUIDED POLICY OPTIMIZATION

Building on the preliminaries, we seek to leverage the insight that Newton's method couples gradients with curvature information—a property that can be particularly valuable in multi-domain RL for LLMs, where interactions between domains are often complex and interdependent. Rather than directly approximating the Newton update, which would be computationally prohibitive in our setting, we distill its essential idea into a lightweight mechanism that induces cross-domain gradient-curvature interactions via sequential task updates. Our method unfolds in three parts: Section 3.1 motivates the design by analyzing the structure of the Newton update, Section 3.2 presents a simple perturbation-based procedure to capture the desired interactions, and Section 3.3 integrates these components into a practical algorithm, i.e., our proposed CGPO. An overview of CGPO is illustrated in Figure 1.

### 3.1 MOTIVATION: WHY HESSIAN-GRADIENT INTERACTIONS MATTER

The starting point of CGPO is an informal observation about Newton's method. Although exact second-order updates are infeasible in large-scale RL for LLMs, the Newton term $\mathbf{Hg}$ (omitting $\theta_t$) couples gradient and curvature, suggesting that such interactions may help reconcile conflicting gradients in multi-domain learning. To illustrate, consider a heuristic expansion: $\mathbf{H}^{-1}\mathbf{g} \approx (\mathbf{I} - (\mathbf{I} - \mathbf{H}))^{-1}\mathbf{g} \approx (\mathbf{I} + (\mathbf{I}-\mathbf{H}) + \mathcal{O}((\mathbf{I}-\mathbf{H})^2))\mathbf{g} \approx 2\mathbf{g} - \mathbf{Hg} + \mathcal{O}((\mathbf{I}-\mathbf{H})^2\mathbf{g})$, where the approximations are informal and serve to reveal the structure rather than provide a rigorous formula. In the multi-domain setting, where $\mathbf{g} = \sum_{k=1}^{K} \mathbf{g}_k$ and $\mathbf{H} = \sum_{k=1}^{K} \mathbf{H}_k$, the product $-\mathbf{Hg}$ then contains cross-domain terms $-\mathbf{H}_j\mathbf{g}_i$ ($i \neq j$), in which the curvature of domain $j$ modulates the gradient of domain $i$.

These interactions effectively transmit curvature signals across tasks, amplifying, dampening, or redirecting updates—capabilities absent in first-order methods. This motivates our key design principle: instead of computing Hessians explicitly, we seek tractable mechanisms that induce such cross-domain interactions to better align multi-domain optimization.

### 3.2 APPROXIMATE CROSS-DOMAIN INTERACTIONS VIA SEQUENTIAL UPDATES

Given the motivation above, the question is how to induce Hessian-gradient interactions without explicitly computing Hessians. Our key idea is to approximate them by observing how the gradient of one domain changes after parameter updates from another.

---

**Algorithm 1** CGPO (one epoch illustration)

---

1: **Input:** $\pi_{\theta_{\text{init}}}$, reward functions $\{R_k\}_{k=1}^K$, datasets $\{D_k\}_{k=1}^K$
2: **Hyperparameter:** number of steps $T, M$, learning rate $\eta$, mixing coefficient $\alpha$
3: **Initialization:** $\pi_{\text{ref}} \leftarrow \pi_{\theta_{\text{init}}}, \pi_{\theta_{\text{new}}} \leftarrow \pi_{\theta_{\text{init}}}$
4: **for** $t = 1, \ldots, T$ **do**
5:     $\pi_{\theta_{\text{old}}} \leftarrow \pi_{\theta_{\text{new}}}$
6:     Sample a batch $D_{(t),k} = \left\{ \mathbf{x}_{(t),k}^{(i)} \right\}_{i=1}^{|D_{(t),k}|}$ from $D_k$ for $1 \leq k \leq K$
7:     Generate responses $\left\{ \mathbf{y}_{(t),k}^{(i,j)} \right\}_{j=1}^G \sim \pi_{\theta_{\text{old}}} \left( \cdot \mid \mathbf{x}_{(t),k}^{(i)} \right)$ for $1 \leq i \leq |D_{(t),k}|, \ 1 \leq k \leq K$
8:     Compute rewards $\left\{ r_{(t),k}^{(i,j)} \right\}_{j=1}^G$ and advantages $\left\{ \hat{A}_{(t),k}^{(i,j)} \right\}_{j=1}^G$ for $1 \leq i \leq |D_{(t),k}|, \ 1 \leq k \leq K$
9:     **for** $m = 1, \ldots, M$ **do**
10:       Sample a mini-batch $D_{(t,m),k}$ from $D_{(t),k}$ for $1 \leq k \leq K$
11:       Let $\sigma(1), \ldots, \sigma(K)$ denote a random permutation of $1, \ldots, K$
12:       $\phi_0 \leftarrow \theta_{\text{new}}$
13:       **for** $k = 1, \ldots, K$ **do**
14:         Update parameters by maximizing Eq. (1) with $D_{(t,m),\sigma(k)}$ and associated responses:

$$\phi_k = \phi_{k-1} - \eta \cdot \frac{|D_{(t,m),\sigma(k)}|}{\sum_{k=1}^K |D_{(t,m),k}|} \cdot \mathbf{g}_{\text{GRPO}} \left( \phi_{k-1}; \theta_{\text{old}}, D_{(t,m),\sigma(k)} \right)$$

15:       $\theta_{\text{new}} \leftarrow \phi_0 + \alpha(\phi_K - \phi_0)$
16: **Output:** $\pi_{\theta_{\text{new}}}$

---

Consider two domains $i$ and $j$. Let domain $i$ updates the parameters from $\theta_{\text{pre}}^{(i)}$ to $\theta_{\text{post}}^{(i)}$. Denoting the Hessian of domain $j$ at $\theta_{\text{pre}}^{(i)}$ by $\mathbf{H}_j \left( \theta_{\text{pre}}^{(i)} \right)$, the gradient of domain $j$ then shifts as

$$\mathbf{g}_j \left( \theta_{\text{post}}^{(i)} \right) - \mathbf{g}_j \left( \theta_{\text{pre}}^{(i)} \right) \approx \mathbf{H}_j \left( \theta_{\text{pre}}^{(i)} \right) \left( \theta_{\text{post}}^{(i)} - \theta_{\text{pre}}^{(i)} \right) \approx \eta \mathbf{H}_j \left( \theta_{\text{pre}}^{(i)} \right) \mathbf{g}_i \left( \theta_{\text{pre}}^{(i)} \right), \qquad (2)$$

which corresponds to the cross-domain product $\mathbf{H}_j \mathbf{g}_i$. This approximation is derived from a first-order Taylor expansion and policy gradient ascent (see Appendix B.1 for the detailed derivation). Thus, sequential updates naturally generate the desired interaction term. Further, to extend beyond two domains, we randomize the order of domains at each iteration. Over time, this exposes every pair of domains to such interactions, allowing curvature information to propagate across domains. Intuitively, each domain *feels* the curvature of others: one nudges the parameters, another responds, producing coordinated updates that help reconcile conflicting objectives.

### 3.3 FULL ALGORITHM: RANDOMIZED CROSS-TASK INTERACTIONS

Building on the insights above, we now introduce CGPO, a principled algorithm for multi-domain policy optimization, illustrated in Figure 1, with pseudocode in Alg. 1. At each training step, we sample batches from all domains and generate multiple candidate responses under the current policy (Lines 6-7). These responses are evaluated by domain-specific reward functions to obtain rewards and advantage estimates (Line 8). We then repeatedly draw mini-batches (Lines 9-10) and perform a randomized sequential update: domains are visited according to a random permutation (Lines 11-13), and at each step the parameters are updated with respect to one domain, conditioned on perturbations induced by previously visited domains (Line 14). Finally, the updated parameters are interpolated with the original ones using a mixing coefficient $\alpha$ (Line 15), stabilizing training by balancing curvature-informed exploration with retention of the base policy.

To understand how sequential updates induce cross-domain Hessian–gradient interactions, consider Lines 11–15. Let the domain order be $\sigma(1), \ldots, \sigma(K)$, and denote the loss, gradient, and Hessian of domain $k$ at parameter $\phi$ by $L_k(\phi)$, $\mathbf{g}_k(\phi)$, and $\mathbf{H}_k(\phi)$. With $\phi_0 \to \phi_1 \to \cdots \to \phi_K$, the gradient of

domain $\sigma(k)$ at $\phi_{k-1}$ can be expanded (see Appendix B.2) as

$$\mathbf{g}_{\sigma(k)}(\phi_{k-1}) = \mathbf{g}_{\sigma(k)}(\phi_0) - \sum_{l=1}^{k-1} \frac{\eta|D_{\sigma(l)}|}{\sum_{s=1}^{K}|D_{\sigma(s)}|} \mathbf{H}_{\sigma(k)}(\phi_0)\mathbf{g}_{\sigma(l)}(\phi_0) + \mathcal{O}(\eta^2). \tag{3}$$

For simplicity, assume uniform batch sizes $|D_{\sigma(l)}|/\sum_{s=1}^{K}|D_{\sigma(s)}| = 1/K$, then

$$\mathbf{g}_{\sigma(k)}(\phi_{k-1}) = \mathbf{g}_{\sigma(k)}(\phi_0) - \frac{\eta}{K} \sum_{l=1}^{k-1} \mathbf{H}_{\sigma(k)}(\phi_0)\mathbf{g}_{\sigma(l)}(\phi_0) + \mathcal{O}(\eta^2). \tag{4}$$

Aggregating over $k$, the overall parameter change after one sequential pass is (see Appendix B.3)

$$\alpha(\phi_K - \phi_0) = -\frac{\alpha\eta}{K} \sum_{k=1}^{K} \mathbf{g}_k(\phi_0) + \frac{\alpha\eta^2}{K^2} \sum_{k=1}^{K} \sum_{l=1}^{k-1} \mathbf{H}_{\sigma(k)}(\phi_0)\mathbf{g}_{\sigma(l)}(\phi_0) + \mathcal{O}(\eta^2). \tag{5}$$

The first term is the aggregated gradient; the second term contains cross-domain Hessian–gradient products. Importantly, the expression above describes the update for a fixed permutation $\sigma$. Because our algorithm re-samples $\sigma$ independently at every iteration, the quantity relevant for understanding CGPO's behavior is the expectation over the random permutation $\sigma$. Taking expectation over $\sigma$ makes every ordered pair $(i, j)$ appear with equal probability; symmetrizing their contributions then yields $\mathbf{H}_i(\phi_0)\mathbf{g}_j(\phi_0) + \mathbf{H}_j(\phi_0)\mathbf{g}_i(\phi_0) = \frac{\partial}{\partial\phi_0}\left(\mathbf{g}_i(\phi_0)^\top \mathbf{g}_j(\phi_0)\right)$ (please see Appendix B.4 for details). This shows that the update encourages alignment of domain gradients. For an analysis of why joint learning does not induce the same cross-domain effect, please see Appendix D.3.

After illustrating how the parameter change encodes both aggregated gradients and cross-domain interactions, it is helpful to clarify the role of the final interpolation step. The vector $\phi_K - \phi_0$ provides a geometry-informed update direction enriched by these interactions. The mixing coefficient $\alpha$ then controls how far we move along this direction (for ablations, see Section 4.3): a sufficiently large $\alpha$ enables the method to benefit from curvature-informed coordination across domains, whereas an excessively large value may push the update outside the locally smooth region where gradient-based approximations remain reliable, potentially destabilizing training—analogous to taking an overly large learning rate in standard optimization. Conversely, setting $\alpha$ too small would under-utilize the information encoded in $\phi_K - \phi_0$ and collapse the update to a near-identity update, losing the benefits introduced sequential interactions. The interpolation therefore functions as a principled mechanism that balances stability and effective use of cross-domain geometric information.

Crucially, this analysis is not restricted to surrogate losses $L_k$: as argued in Section 2.2, GRPO surrogates provide faithful approximations of the true policy gradients within their trust regions. Thus, the induced interactions improve alignment not only among surrogate gradients but also among the true policy gradients $\nabla_\theta \mathcal{J}_k(\theta)$. In effect, randomized sequential updates encourage cooperation across domains by introducing curvature–gradient couplings that steer optimization toward coordinated improvements on the full multi-domain objective $\sum_{k=1}^{K} \mathcal{J}_k(\theta)$.

**Discussion.** We highlight two clarifications to better situate our approach.

- Sequential updates is a common technique across different learning paradigms. For example, in meta-learning, Reptile (Nichol et al., 2018) adopts sequential updates to learn an initial model for rapid adaptation to new tasks, while in federated learning, methods such as FedAvg (McMahan et al., 2017) aggregate sequential client updates to improve global optimization. However, these precedents do not diminish the novelty of our contributions. First, our sequential update originates from our observation of Newton's method and its capability to navigate complex landscapes, where inherent curvature–gradient interactions naturally emerge across domains. Second, we adapt this mechanism to the multi-domain RL for LLMs setting, where domain-specific rewards and surrogate policy gradients pose unique challenges absent in meta-learning or federated learning. Finally, we integrate randomized ordering, surrogate faithfulness (via GRPO), and stabilization through interpolation into a unified algorithm tailored for large-scale RLHF. These innovations collectively distinguish CGPO as a novel and practical solution for multi-domain policy optimization.

- A natural concern is that multiple updates per step could inflate the effective learning rate. To avoid this, we scale each gradient proportionally to its mini-batch size and normalize by the total across domains. This ensures that the overall update magnitude is consistent with that of using a single aggregated batch, thereby preserving comparability with standard mini-batch optimization.

## 4 EXPERIMENTS

### 4.1 EXPERIMENTAL SETTINGS

**Tasks and Datasets.** We focus on enhancing the LLMs' overall capabilities across four domains—mathematical reasoning, code generation, scientific QA, and creative writing. These domains not only represent **core areas of current research interest** but also **span four distinct forms of reward feedback**, thereby ensuring both **comprehensiveness** and **diversity**. For mathematics, code, and science, we construct subsets from the Guru dataset (Cheng et al., 2025) with attention to dataset size and sample difficulty (as Guru poses non-trivial challenges for 7B-scale models): the math subset contains 6,250 samples, consisting of the 5,000 easiest problems (ranked by the pass rate of Qwen2.5-7B-Instruct) and 1,250 more challenging ones; the code subset totals 4,740 samples, comprising all 3,791 problems with a Qwen2.5-7B-Instruct's pass rate of at least 25% plus an additional 949 randomly sampled from the remainder, ensuring a roughly 4:1 ratio between easier and harder samples; and the scientific QA subset includes the entire STEM split of Guru, with 3,591 samples. For creative writing, we randomly sample 2,000 samples each from the three most popular datasets available on Huggingface (LitBench (Fein et al., 2025), Creative_Writing-ShareGPT (Nitral-AI, 2024), and wildchat-creative-writing-3k-rft (kevinshin, 2025)), yielding a dataset of 6,000 samples. For details of the datasets, please see Appendix C.1.

**Baselines.** We compare our CGPO with several representative baselines. For vanilla strategies, we include joint learning, which directly trains on a multi-domain dataset without any special strategies. For curriculum learning (CL), following the taxonomy in (Soviany et al., 2022), we include Omni-Thinker (Li et al., 2025a), a *progressive CL* method, and *self-paced CL*, which schedules training from easier to harder examples based on task difficulty (measured by pass rate). For gradient balancing, we include FAMO (Liu et al., 2023), categorized in (Chen et al., 2025b) as a representative approach for balancing gradient magnitudes across domains. We also attempted to implement gradient manipulation methods such as PCGrad (Yu et al., 2020), but these require simultaneously storing and operating on multiple per-domain gradients on devices, which leads to out-of-memory (OOM) issues in the RL for LLM setting. For more details of baselines, please refer to Appendix C.2.

**Training Details.** We train Qwen2.5-3B-Instruct and Qwen2.5-7B-Instruct on the multi-domain dataset using the verl framework (Sheng et al., 2025). For the implementation of multi-domain training in terms of data processing and reward design, we follow the codebases of (Cheng et al., 2025) and (Ma et al., 2025). For math, we adopt rule-based rewards; for coding, we evaluate models' outputs using unit test cases based on SandboxFusion (Bytedance-Seed-Foundation-Code-Team et al., 2025); for scientific QA, we use a 1.5B General-Verifier (Ma et al., 2025) to assess the consistency between model outputs and groundtruth answers; and for creative writing, we compare model responses with reference answers using Qwen2.5-72B-Instruct. Besides, we require the model to enclose its reasoning process within `<think></think>` tags and penalize responses that violate this format requirement, along with domain-specific constraints. Details of the reward functions are provided in Appendix C.3. We use a learning rate of $1 \times 10^{-6}$, a prompt batch size of $128$, a mini-batch size of $64$, a group size of $8$, a rollout temperature of $1.0$, $\varepsilon_{low} = 0.2$, $\varepsilon_{high} = 0.28$, and $\beta = 0.001$ for CGPO and all baselines. Our code is available at `https://github.com/MIRALab-USTC/CGPO`. For more details of hyperparameters, please see Appendix C.4.

**Evaluation.** We evaluate our models on seven widely-used benchmarks: MATH500 (Hendrycks et al., 2021), AMC 2023 (MAA, 2023), HumanEval (Chen et al., 2021), MBPP (Austin et al., 2021), GPQA-diamond (Rein et al., 2023), SuperGPQA (Team et al., 2025), and WritingBench (Wu et al., 2025). To ensure consistent scaling across benchmarks, the scores on WritingBench are multiplied by 10. We use vLLM (Kwon et al., 2023) for efficient inference, generating 4 responses per query with a temperature of 0.6 and top-$p$ sampling of 0.95. Further details can be found in Appendix C.5.

### 4.2 MAIN RESULTS

**CGPO boosts the multi-domain reasoning of LLMs.** Table 1 presents the results across different methods. From the table we make the following observations: (1) CGPO achieves the **highest average performance** for both model scales (3B and 7B), ranking either first or second in most individual domains. This demonstrates its effectiveness in enhancing multi-domain reasoning capabilities of LLMs. (2) For smaller models (3B), CGPO consistently outperforms other baselines on *code generation* and *creative writing*, while maintaining competitive performance on *math* and *scientific QA*. FAMO and Omni-Thinker also provide gains over joint learning, particularly in *code generation*

Table 1: **Performance of models (Qwen2.5-3B-Instruct and Qwen2.5-7B-Instruct) trained on the multi-domain dataset with different methods, evaluated on multiple benchmarks.** The bold font indicates the best result and an underline indicates the second-best result.

| Methods | Math | | Code Generation | | Scientific QA | | Creative Writing | AVG |
|---|---|---|---|---|---|---|---|---|
| | MATH500 | AMC | HumanEval | MBPP | GPQA-diamond | SuperGPQA | WritingBench | |
| *# Qwen2.5-3B-Instruct* | | | | | | | | |
| Joint Learning | 64.50 | 39.38 | 72.39 | 59.40 | **24.87** | 24.12 | 58.61 | *49.04* |
| Omni-Thinker | **65.65** | **41.50** | 71.95 | 58.80 | 21.34 | **26.75** | 57.90 | *49.13* |
| Self-paced CL | 65.30 | 38.75 | 70.12 | 58.80 | 24.37 | 24.72 | 57.82 | *48.55* |
| FAMO | 63.80 | 39.12 | 72.48 | 59.20 | 23.47 | 26.51 | 58.46 | *49.01* |
| **CGPO** | 64.20 | 39.71 | **74.29** | **60.80** | 24.37 | 26.63 | **63.04** | *★50.42* |
| *# Qwen2.5-7B-Instruct* | | | | | | | | |
| Joint Learning | **76.00** | 56.25 | 79.88 | 68.60 | 19.70 | **32.75** | 63.15 | *56.62* |
| Omni-Thinker | 75.10 | 53.75 | 82.93 | 68.60 | 23.86 | 30.63 | 62.35 | *56.75* |
| Self-paced CL | 74.70 | 51.88 | 82.93 | 68.00 | 21.72 | 30.25 | 63.68 | *56.17* |
| FAMO | 75.65 | 55.63 | 82.54 | 68.80 | 23.07 | 31.49 | 63.62 | *57.26* |
| **CGPO** | 75.55 | **59.38** | **84.15** | **72.00** | **26.77** | **32.75** | **66.52** | *★59.59* |

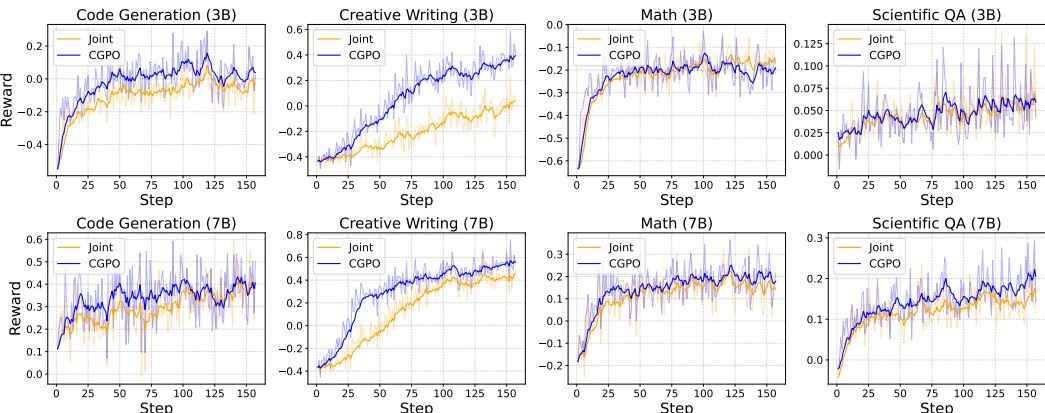

Figure 2: Training reward curves for Qwen2.5-3B-Instruct and Qwen2.5-7B-Instruct on four domains (code, creative writing, math, and scientific QA), comparing CGPO and joint learning.

and *scientific QA*, but they lag behind CGPO in *creative writing*. Self-paced CL remains the weakest overall, likely due to imbalanced domain difficulty and insufficient coverage of informative responses at different training stages. (3) For larger models (7B), CGPO achieves **clear improvements across nearly all domains**, with the largest gains on *code generation* and *creative writing*, highlighting that its benefits scale with model capacity. Notably, FAMO shows competitive results, especially in *math* and *creative writing*, confirming that gradient balancing can help, but it still falls short of CGPO in aggregating multi-domain knowledge effectively. These results collectively indicate that curriculum learning and gradient weighting methods can provide partial improvements, but their reliance on task difficulty, loss, or gradient magnitude alone is insufficient. In contrast, CGPO leverages geometric information via randomized sequential updates and interpolation, enabling coordinated multi-domain optimization and consistent performance gains across mathematical reasoning, code generation, scientific QA, and open-ended creative tasks.

**CGPO achieves faster reward improvement across all domains.** Figure 2 presents the training reward curves of Qwen2.5-3B-Instruct and Qwen2.5-7B-Instruct on the four domains, with all curves smoothed using EMA to clearly reveal trends. For both model sizes, the curves of CGPO consistently remain above those of joint learning. The advantage is particularly pronounced in *code generation* and *creative writing*, while in *math* and *scientific QA* the improvement is evident but less striking. Notably, compared with the other three domains, *creative writing* is more subjective, requiring the model to generate diverse and creative outputs rather than strictly structured or precise answers; this makes **potential conflicts with the other domains the largest**. The substantial advantage of CGPO

Table 3: **Ablation study on domain order randomization in CGPO with Qwen2.5-7B-Instruct.** The bold font indicates the better result.

| Methods | Math | | Code Generation | | Scientific QA | | Creative Writing | AVG |
|---|---|---|---|---|---|---|---|---|
| | MATH500 | AMC | HumanEval | MBPP | GPQA-diamond | SuperGPQA | WritingBench | |
| CGPO_fix | **77.20** | 56.88 | 83.54 | 69.60 | 23.08 | 31.75 | **67.30** | *58.48* |
| CGPO | 75.55 | **59.38** | **84.15** | **72.00** | **26.77** | **32.75** | 66.52 | *59.59* |

Table 4: **Ablation study on the effect of the mixing coefficient $\alpha$ in CGPO with Qwen2.5-7B-Instruct.** The bold font indicates the best result and an underline indicates the second-best result.

| $\alpha$ | Math | | Code Generation | | Scientific QA | | Creative Writing | AVG |
|---|---|---|---|---|---|---|---|---|
| | MATH500 | AMC | HumanEval | MBPP | GPQA-diamond | SuperGPQA | WritingBench | |
| 0.9 | **75.85** | 55.88 | **84.15** | 71.20 | 21.72 | 32.25 | 66.01 | *58.15* |
| 1.2 | 75.55 | **59.38** | **84.15** | **72.00** | **26.77** | 32.75 | **66.52** | *59.59* |
| 1.5 | 75.55 | 55.25 | 81.10 | 69.20 | 23.36 | **35.37** | 66.47 | *58.04* |

in the reward curve for *creative writing* compared to joint learning provides **strong evidence that CGPO effectively mitigates cross-domain conflicts**. We also observe considerable differences in initial reward levels across domains. Taking Qwen2.5-7B-Instruct as an example, *creative writing* and *scientific QA* start near $-0.4$ and 0, respectively, reflecting largely incorrect outputs, whereas *math* and especially *coding* begin from higher baselines (coding around 0.1). This indicates that the models enter RL training with uneven domain-specific capabilities. Importantly, CGPO delivers varing degrees of acceleration even for domains with comparable starting points, suggesting that factors such as dataset difficulty or reward function design may influence the speedup. Investigating the underlying causes of these differences is left for future work.

## 4.3 ANALYSIS AND ABLATIONS

**CGPO introduces only negligible additional computation overhead.** In multi-domain RL for LLMs, the dominant computational bottleneck typically lies in generating responses and computing rewards—particularly in domains such as *coding* and *creative writing*—rather than in the forward and backward passes of the model itself. Against this backdrop, the additional operations introduced by CGPO are minimal. The sequential updates across domains are essentially equivalent to splitting a mini-batch into smaller chunks and processing them sequentially, which incurs almost the same computational cost as standard mini-batch training. Furthermore, the final interpolation with the mixing coefficient $\alpha$ amounts

Table 2: **Computation cost comparison between joint learning and CGPO (1 epoch).** Note that the units of total time and per-step time are different (hours vs. minutes).

| Methods | Total (h) | Step (min) |
|---|---|---|
| *# Qwen2.5-3B-Instruct* | | |
| Joint Learning | 14.8 | 5.58 |
| CGPO | 16.0 | 6.04 |
| *# Qwen2.5-7B-Instruct* | | |
| Joint Learning | 17.8 | 6.72 |
| CGPO | 18.6 | 7.02 |

to a single vector operation, which is computationally negligible. Taken together, these factors ensure that the overall overhead of CGPO is practically insignificant, and the total training cost remains nearly identical to that of joint learning. As shown in Table 2, the per-step wall-clock time under CGPO is only slightly higher than joint learning, confirming that our method adds no meaningful overhead in practice. For timing experiments on 32B and 72B models, please see Appendix E.1.

**Randomizing domain order is necessary for effective cross-domain interactions.** We conduct ablations to examine the necessity of randomizing domain order. Specifically, we compare the standard randomized variant with a fixed-order variant (CGPO_fix), where the sequence of domains remains unchanged throughout training. As shown in Table 3, randomizing the order consistently leads to higher average performance across all benchmarks. This result highlights that randomization is essential: it ensures balanced sequential updates among domains, avoiding systematic bias in Hessian–gradient interactions. In contrast, fixed ordering allows earlier domains to dominate updates, while later domains can only adapt passively, reducing overall multi-domain coordination.

**The mixing coefficient $\alpha$ plays a critical role in balancing stability and curvature exploitation.** To study its effect, we experiment with $\alpha \in \{0.9, 1.2, 1.5\}$ and report the corresponding multi-domain performance in Table 4. Among these choices, $\alpha = 1.2$ achieves the best overall average, reflecting a favorable trade-off between retaining the base policy and incorporating curvature-informed updates. Notably, the average performance of all tested $\alpha$ values exceeds that of the strongest baseline, FAMO (57.26), indicating that CGPO is robust to the choice of $\alpha$. The fact that all $\alpha$ values are close to 1.0 suggests that the interpolation does not substantially change the effective learning rate; the observed gains therefore arise from the curvature-aware sequential updates rather than step size adjustments.

## 5 RELATED WORK

**Multi-domain RL for LLMs.** It is challenging to stably improve LLMs' multi-domain reasoning performance with RL. One difficulty lies in designing reward functions that generalize across diverse tasks. Some works propose broadly applicable reward computation, such as simplifying binary rewards via ground-truth properties (Zhou et al., 2025) or using correctness likelihood (Yu et al., 2025b). Others adopt domain-specific reward designs, e.g., hybrid rule-based, sandbox, and LLM-as-a-judge systems (Li et al., 2025a). A second challenge is understanding cross-domain interactions: Cheng et al. (2025) analyze how single-domain training affects other skills, while Li et al. (2025b) extend this to math, coding, and puzzles. TAPPA (Yang et al., 2026) reveals consistent properties of LLM attention patterns via query self-similarity across multiple tasks, but it is primarily diagnostic and offers no optimization mechanism for mitigating cross-domain interference in multi-domain RL. EVIC (Liang et al., 2025) effectively improves multi-domain SFT for LLMs via curriculum learning guided by evolving gradient-based interactions; yet it does not readily extend to RL, since responses are generated online and the corresponding gradients cannot be precomputed in advance.

**Mitigating Gradient Conflicts.** Gradient interference is a major obstacle in multi-task learning (Chen et al., 2025b). Approaches such as GradNorm (Chen et al., 2018), PCGrad (Yu et al., 2020), MGDA (Sener & Koltun, 2018), ConFIG (Liu et al., 2024), and CAGrad (Liu et al., 2021) resolve conflicts by balancing or projecting task gradients. While effective in standard MTL settings, they are difficult to scale to RL for LLMs: many require storing all task gradients on-device, causing memory bottlenecks, or operate reactively without leveraging reward-landscape geometry, leading to high variance under noisy rollout-based gradients. These limitations motivate scalable, memory-efficient mechanisms for mitigating cross-domain conflicts, as pursued by CGPO.

**Second-Order Optimization Methods.** The loss landscapes of deep neural networks are often highly complex, posing challenges for first-order optimization algorithms, such as gradient descent. Without insights into the geometric structure of the landscape, first-order methods can easily get trapped in saddle points or narrow valleys, making it difficult to reach better local optima. In contrast, second-order optimization methods, such as Newton's method, exploit geometric information like the Hessian matrix to precondition gradients according to the local curvature, offering stronger theoretical guarantees. To mitigate the computational cost of full Hessian computation, various approximate Newton methods have been proposed, including AdaGrad (Duchi et al., 2011), K-FAC (Martens & Grosse, 2015), GGT (Agarwal et al., 2018), Shampoo (Gupta et al., 2018), and SOAP (Vyas et al., 2025). Recent studies show that Newton's method and SOAP (Vyas et al., 2025) can alleviate gradient conflicts in PINNs (Wang et al., 2025), providing inspiration for our approach. However, due to the massive parameter scale of LLMs, directly applying approximate variants of Newton's method in RL for LLMs is fundamentally infeasible (we provide a detailed discussion in Appendix D.4). Motivated by this, we distill the core idea of leveraging curvature information and develop CGPO, a principled and scalable framework for multi-domain RL in LLMs.

## 6 CONCLUSION AND LIMITATIONS

We present CGPO, a principled and scalable framework for multi-domain RL of LLMs. Inspired by Newton's method, CGPO leverages the geometric structure of the reward surfaces to precondition gradients, while avoiding the cost of full Hessian computation. Through randomized sequential updates, each domain's gradient is modulated by curvature information from other domains, fostering cross-domain interactions and implicitly aligning gradients. Experiments on a diverse multi-domain dataset covering mathematical reasoning, code generation, scientific QA, and creative writing show that CGPO outperforms all baselines, achieving faster reward improvement and stronger multi-domain reasoning across all benchmarks. For the limitations of this paper, please refer to Appendix F.

## 7 ETHICS STATEMENT

This work studies multi-domain reinforcement learning for LLMs using publicly available or appropriately licensed datasets across domains such as mathematics, coding, scientific QA, and creative writing. No human subjects were directly involved. While our methods improve cross-domain optimization, models trained with them could be misused to produce plausible but incorrect or unsafe outputs. We strongly discourage any deployment outside research contexts and emphasize that reward functions and training setups are designed to encourage safe and aligned outputs. All research was conducted in accordance with the ICLR Code of Ethics, with no conflicts of interest or external influence on methodology or results.

## 8 REPRODUCIBILITY STATEMENT

To facilitate reproducibility, we provide detailed descriptions of our algorithm (CGPO) in Section 3.3 and Algorithm 1, including pseudo-code and key hyperparameters. Experimental setups, including data processing, reward functions, and evaluation benchmarks, are described in Section 4 and Appendix C. Where applicable, we provide references to publicly available datasets. Our code is available at `https://github.com/MIRALab-USTC/CGPO`. All derivations, approximations, and additional analyses supporting the method are included in Appendix B. Together, these materials provide sufficient information for replication of the reported results.

### ACKNOWLEDGMENTS

This work was supported in part by National Key R&D Program of China under contract 2022ZD0119801, National Nature Science Foundations of China grants U23A20388 and 62021001. We would like to thank all the anonymous reviewers for their insightful comments.

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

## A LLM USAGE STATEMENT

In preparing this manuscript, we used a large language model (LLM) in two distinct ways. First, we employed LLMs as an assistive tool for text refinement, including improving grammar, wording, and clarity. Second, LLMs themselves are the primary subject of this research: we study reinforcement learning (RL) training for LLMs. Accordingly, all experiments involve using large models for training, inference, and scoring, as part of the methodology under investigation.

All scientific content, including problem formulation, methodology, experiments, and conclusions, was developed and verified entirely by the authors. The authors take full responsibility for the integrity and accuracy of the manuscript. No LLM was credited as an author, and all substantive research contributions are attributable exclusively to the human authors.

## B MATHEMATICAL DERIVATIONS

### B.1 DETAILED DERIVATION OF EQ. (2)

Eq. (2) in Section 3.2 states:

$$\mathbf{g}_j\left(\theta_{\text{post}}^{(i)}\right) - \mathbf{g}_j\left(\theta_{\text{pre}}^{(i)}\right) \approx \mathbf{H}_j\left(\theta_{\text{pre}}^{(i)}\right)\left(\theta_{\text{post}}^{(i)} - \theta_{\text{pre}}^{(i)}\right) \approx \eta\mathbf{H}_j\left(\theta_{\text{pre}}^{(i)}\right)\mathbf{g}_i\left(\theta_{\text{pre}}^{(i)}\right). \tag{6}$$

**Derivation:** Assuming the gradient function $\mathbf{g}_j(\theta)$ is smooth, we apply a first-order Taylor expansion around $\theta_{\text{pre}}^{(i)}$:

$$\mathbf{g}_j\left(\theta_{\text{post}}^{(i)}\right) \approx \mathbf{g}_j\left(\theta_{\text{pre}}^{(i)}\right) + \mathbf{H}_j\left(\theta_{\text{pre}}^{(i)}\right)\left(\theta_{\text{post}}^{(i)} - \theta_{\text{pre}}^{(i)}\right) + \mathcal{O}(\|\Delta\theta\|^2), \tag{7}$$

where $\mathbf{H}_j(\theta) = \nabla_\theta^2 L_j(\theta)$ is the Hessian matrix for domain $j$, and $\Delta\theta = \theta_{\text{post}}^{(i)} - \theta_{\text{pre}}^{(i)}$. Neglecting higher-order terms and rearranging gives:

$$\mathbf{g}_j\left(\theta_{\text{post}}^{(i)}\right) - \mathbf{g}_j\left(\theta_{\text{pre}}^{(i)}\right) \approx \mathbf{H}_j\left(\theta_{\text{pre}}^{(i)}\right)\left(\theta_{\text{post}}^{(i)} - \theta_{\text{pre}}^{(i)}\right). \tag{8}$$

In policy optimization, parameters are updated via gradient ascent (maximizing rewards):

$$\theta_{\text{post}}^{(i)} = \theta_{\text{pre}}^{(i)} + \eta\mathbf{g}_i\left(\theta_{\text{pre}}^{(i)}\right), \tag{9}$$

where $\eta$ is the learning rate. Substituting this into the previous equation yields:

$$\theta_{\text{post}}^{(i)} - \theta_{\text{pre}}^{(i)} = \eta\mathbf{g}_i\left(\theta_{\text{pre}}^{(i)}\right), \tag{10}$$

and therefore,

$$\mathbf{g}_j\left(\theta_{\text{post}}^{(i)}\right) - \mathbf{g}_j\left(\theta_{\text{pre}}^{(i)}\right) \approx \eta\mathbf{H}_j\left(\theta_{\text{pre}}^{(i)}\right)\mathbf{g}_i\left(\theta_{\text{pre}}^{(i)}\right), \tag{11}$$

which is Eq. (2). This approximation shows that the gradient update from domain $i$ influences the gradient of domain $j$ through the curvature of domain $j$.

### B.2 DETAILED DERIVATION OF EQ. (3) AND EQ. (4)

Eq. (3) and Eq. (4) in Section 3.3 state:

$$\mathbf{g}_{\sigma(k)}(\phi_{k-1}) = \mathbf{g}_{\sigma(k)}(\phi_0) - \sum_{l=1}^{k-1}\frac{\eta|D_{\sigma(l)}|}{\sum_{s=1}^{K}|D_{\sigma(s)}|}\mathbf{H}_{\sigma(k)}(\phi_0)\mathbf{g}_{\sigma(l)}(\phi_0) + \mathcal{O}(\eta^2) \tag{12}$$

$$\mathbf{g}_{\sigma(k)}(\phi_{k-1}) = \mathbf{g}_{\sigma(k)}(\phi_0) - \frac{\eta}{K}\sum_{l=1}^{k-1}\mathbf{H}_{\sigma(k)}(\phi_0)\mathbf{g}_{\sigma(l)}(\phi_0) + \mathcal{O}(\eta^2) \tag{13}$$

**Derivation:** Consider the randomized sequential update: domains are processed in the order $\sigma(1), \ldots, \sigma(K)$. The parameter update for each domain (using gradient ascent) is:

$$\phi_k = \phi_{k-1} + \eta_k\mathbf{g}_{\sigma(k)}(\phi_{k-1}), \tag{14}$$

where $\eta_k = \eta |D_{\sigma(k)}| / \sum_{s=1}^{K} |D_{\sigma(s)}|$ is the scaled learning rate.

For domain $\sigma(k)$, its gradient is evaluated at $\phi_{k-1}$. Using a Taylor expansion around $\phi_0$:

$$\mathbf{g}_{\sigma(k)}(\phi_{k-1}) = \mathbf{g}_{\sigma(k)}(\phi_0) + \mathbf{H}_{\sigma(k)}(\phi_0)(\phi_{k-1} - \phi_0) + \mathcal{O}(\eta^2). \tag{15}$$

Now compute $\phi_{k-1} - \phi_0$. Note that:

$$\phi_{k-1} = \phi_0 + \sum_{l=1}^{k-1}(\phi_l - \phi_{l-1}) = \phi_0 + \sum_{l=1}^{k-1} \eta_l \mathbf{g}_{\sigma(l)}(\phi_{l-1}). \tag{16}$$

To first order, we approximate $\mathbf{g}_{\sigma(l)}(\phi_{l-1}) \approx \mathbf{g}_{\sigma(l)}(\phi_0)$ (error $\mathcal{O}(\eta^2)$):

$$\phi_{k-1} - \phi_0 \approx \sum_{l=1}^{k-1} \eta_l \mathbf{g}_{\sigma(l)}(\phi_0). \tag{17}$$

Substituting into the Taylor expansion:

$$\mathbf{g}_{\sigma(k)}(\phi_{k-1}) \approx \mathbf{g}_{\sigma(k)}(\phi_0) + \mathbf{H}_{\sigma(k)}(\phi_0)\left(\sum_{l=1}^{k-1} \eta_l \mathbf{g}_{\sigma(l)}(\phi_0)\right) + \mathcal{O}(\eta^2). \tag{18}$$

Substituting $\eta_l = \eta |D_{\sigma(l)}| / \sum_{s=1}^{K} |D_{\sigma(s)}|$ gives Eq. (3).

If we assume uniform batch sizes, i.e., $|D_{\sigma(l)}| / \sum_{s=1}^{K} |D_{\sigma(s)}| = 1/K$, then $\eta_l = \eta/K$, which simplifies to Eq. (4).

## B.3 DETAILED DERIVATION OF EQ. (5)

Eq. (5) in Section 3.3 states:

$$\alpha(\phi_K - \phi_0) = -\frac{\alpha\eta}{K} \sum_{k=1}^{K} \mathbf{g}_k(\phi_0) + \frac{\alpha\eta^2}{K^2} \sum_{k=1}^{K} \sum_{l=1}^{k-1} \mathbf{H}_{\sigma(k)}(\phi_0)\mathbf{g}_{\sigma(l)}(\phi_0) + \mathcal{O}(\eta^2). \tag{19}$$

**Derivation:** The total parameter change is:

$$\phi_K - \phi_0 = \sum_{k=1}^{K}(\phi_k - \phi_{k-1}) = \sum_{k=1}^{K} \eta_k \mathbf{g}_{\sigma(k)}(\phi_{k-1}). \tag{20}$$

Using the approximation from Eq. (4) (uniform batch sizes):

$$\mathbf{g}_{\sigma(k)}(\phi_{k-1}) \approx \mathbf{g}_{\sigma(k)}(\phi_0) - \frac{\eta}{K} \sum_{l=1}^{k-1} \mathbf{H}_{\sigma(k)}(\phi_0)\mathbf{g}_{\sigma(l)}(\phi_0), \tag{21}$$

and substituting $\eta_k = \eta/K$:

$$\phi_K - \phi_0 \approx \sum_{k=1}^{K} \frac{\eta}{K}\left[\mathbf{g}_{\sigma(k)}(\phi_0) - \frac{\eta}{K} \sum_{l=1}^{k-1} \mathbf{H}_{\sigma(k)}(\phi_0)\mathbf{g}_{\sigma(l)}(\phi_0)\right]$$

$$= \frac{\eta}{K} \sum_{k=1}^{K} \mathbf{g}_{\sigma(k)}(\phi_0) - \frac{\eta^2}{K^2} \sum_{k=1}^{K} \sum_{l=1}^{k-1} \mathbf{H}_{\sigma(k)}(\phi_0)\mathbf{g}_{\sigma(l)}(\phi_0). \tag{22}$$

Multiplying by the mixing coefficient $\alpha$:

$$\alpha(\phi_K - \phi_0) \approx \frac{\alpha\eta}{K} \sum_{k=1}^{K} \mathbf{g}_{\sigma(k)}(\phi_0) - \frac{\alpha\eta^2}{K^2} \sum_{k=1}^{K} \sum_{l=1}^{k-1} \mathbf{H}_{\sigma(k)}(\phi_0)\mathbf{g}_{\sigma(l)}(\phi_0). \tag{23}$$

Note that $\sum_{k=1}^{K} \mathbf{g}_{\sigma(k)}(\phi_0) = \sum_{k=1}^{K} \mathbf{g}_k(\phi_0)$ (permutation invariant), yielding Eq. (5).

## B.4 DERIVATION OF GRADIENT ALIGNMENT SYMMETRIZATION

In Section 3.3, it is mentioned that after randomization, the cross-term expectation symmetrizes as:

$$\mathbf{H}_i(\phi_0)\mathbf{g}_j(\phi_0) + \mathbf{H}_j(\phi_0)\mathbf{g}_i(\phi_0) = \frac{\partial}{\partial \phi_0}\left(\mathbf{g}_i(\phi_0)^\top \mathbf{g}_j(\phi_0)\right). \tag{24}$$

**Derivation:** The key mathematical insight is the following identity concerning the gradient of the inner product between two gradients.

Consider the inner product $S(\phi_0) = \mathbf{g}_i(\phi_0)^\top \mathbf{g}_j(\phi_0)$. The gradient of this scalar function $S$ with respect to $\phi_0$ is given by:

$$\nabla_{\phi_0} S = \nabla_{\phi_0}\left(\mathbf{g}_i(\phi_0)^\top \mathbf{g}_j(\phi_0)\right) = \mathbf{H}_i(\phi_0)\mathbf{g}_j(\phi_0) + \mathbf{H}_j(\phi_0)\mathbf{g}_i(\phi_0), \tag{25}$$

where we have used the product rule and the symmetry of the Hessian matrices, $\mathbf{H}_j = \mathbf{H}_j^\top$. This result can be seen by noting that the derivative of $\mathbf{g}_i^\top \mathbf{g}_j$ w.r.t. $\phi_0$ is $(\partial \mathbf{g}_i/\partial \phi_0)^\top \mathbf{g}_j + \mathbf{g}_i^\top(\partial \mathbf{g}_j/\partial \phi_0) = \mathbf{H}_i \mathbf{g}_j + \mathbf{g}_i^\top \mathbf{H}_j$. Since $\mathbf{g}_i^\top \mathbf{H}_j$ is a row vector, its transpose is $\mathbf{H}_j \mathbf{g}_i$. The gradient (as a column vector) is therefore $\mathbf{H}_i \mathbf{g}_j + \mathbf{H}_j \mathbf{g}_i$.

Under a randomized ordering $\sigma$, the expectation of the cross-term involving $\mathbf{H}_{\sigma(k)}\mathbf{g}_{\sigma(l)}$ for $k > l$ will involve pairs $(i, j)$ symmetrically. The update term derived from the second-order expansion is proportional to $\mathbf{H}_i \mathbf{g}_j$. The symmetric form $\mathbf{H}_i \mathbf{g}_j + \mathbf{H}_j \mathbf{g}_i$ appearing in the gradient of the inner product $\nabla_{\phi_0}(\mathbf{g}_i^\top \mathbf{g}_j)$ indicates that, in expectation, the update encourages an increase in the inner product between the gradients of different domains, thus promoting their alignment.

**Remark.** We would like to clarify the intended meaning of Eq. (5) and the role of the expectation over permutations, in order to avoid possible ambiguities and to keep the presentation self-contained.

**(1) Interpretation of Eq. (5).** Eq. (5) is obtained from a deterministic Taylor expansion of one sequential update pass conditioned on a fixed permutation $\sigma$. The resulting parameter change decomposes into: (i) a first-order term corresponding to aggregated gradients, and (ii) a second-order interaction term involving Hessian-gradient products. These Hessian-gradient interaction terms arise deterministically from executing a sequential update under a specific ordering; they do not rely on randomness or averaging. The expression makes explicit the structural cross-domain second-order interactions induced by sequential updates.

**(2) Role of the expectation over $\sigma$.** The expectation over permutations is used to express a symmetry property. To make this more concrete, imagine that at the same parameter $\theta_t$, we were able—*hypothetically, since the algorithm does not actually do this*—to sample $M$ independent permutations $\left\{\sigma_t^{(m)}\right\}_{m=1}^M$, each corresponding to an ordering $\tau_t^{(m)} = \left(\sigma_t^{(m)}(k)\right)_{k=1}^K$. In this hypothetical scenario, as $M \to \infty$, the events "$i$ appears before $j$" and "$j$ appears before $i$" would occur with essentially equal frequency for every pair $(i, j)$. This limiting symmetry is exactly what our expectation argument is intended to express, and it is what leads to the symmetric combination $\mathbf{H}_i \mathbf{g}_j + \mathbf{H}_j \mathbf{g}_i$ in the discussion following Eq. (5).

In the actual algorithm, of course, we sample **only one** permutation at each iteration. This introduces sampling **error**—but not **bias in the expectation sense**—because we do not average over multiple permutations.

Importantly, this sampling error does not accumulate in a harmful way in practice. A helpful way to view this is through **an analogy with standard SGD**: each stochastic gradient is, in expectation, equal to the true gradient (just as the contributions of $\mathbf{H}_i \mathbf{g}_j$ and $\mathbf{H}_j \mathbf{g}_i$ are symmetric in expectation), yet in practice we use only one stochastic gradient per step rather than averaging many samples—just as our algorithm samples only one permutation per iteration rather than averaging over many permutations at the same parameter. This practice in SGD does introduce variance and error, but it does not undermine either the effectiveness of SGD or the usefulness of the statement that "the stochastic gradient equals the true gradient in expectation". The same phenomenon appears in our algorithm.

Therefore, when we refer to an expectation, we mean the conditional expectation taken at a fixed $\theta_t$, i.e., conditional on the past history $\mathcal{F}_{t-1}$—just as the expectation of a stochastic gradient in SGD is interpreted conditional on the current parameter value.

## C   MORE DETAILS OF EXPERIMENTS

### C.1   TASKS AND DATASETS

We focus on enhancing LLMs' overall capabilities across four domains—mathematical reasoning, code generation, scientific QA, and creative writing. These domains not only represent **core areas of current research interest** but also **span four distinct forms of reward feedback**, thereby ensuring both **comprehensiveness** and **diversity**.

- **Mathematics**: we construct a subset of 6,250 samples from the Guru dataset (Cheng et al., 2025). This includes the 5,000 easiest problems (ranked by the pass rate of Qwen2.5-7B-Instruct) and 1,250 more challenging ones, ensuring a balance between accessible and difficult problems.

- **Code generation**: we select a total of 4,740 samples from Guru. Specifically, we take all 3,791 problems with a Qwen2.5-7B-Instruct's pass rate of at least 25% and add 949 problems randomly sampled from the remainder, yielding an approximate 4:1 ratio between easier and harder samples.

- **Scientific QA**: we include the entire STEM split of Guru, resulting in 3,591 samples. This preserves the full coverage of science-related reasoning tasks while maintaining consistency with prior benchmarks.

- **Creative writing**: we randomly sample 2,000 samples each from three popular Huggingface datasets—LitBench (Fein et al., 2025), Creative_Writing-ShareGPT (Nitral-AI, 2024), and wildchat-creative-writing-3k-rft (kevinshin, 2025)—to construct a dataset of 6,000 samples, ensuring stylistic variety and broad coverage of open-ended writing abilities.

### C.2   BASELINES

We compare our CGPO against four representative baselines: joint learning, Omni-Thinker (Li et al., 2025a), Self-Paced CL, and FAMO (Liu et al., 2023).

- **Joint learning.** Joint learning is the most basic paradigm in MTL. It aggregates the loss functions of all tasks into a single objective, enabling simultaneous optimization. As a straightforward training strategy without any task-specific adjustments, joint learning serves as a reference point for evaluating improvements brought by more advanced methods.

- **Omni-Thinker.** Omni-Thinker belongs to *progressive CL* methods as categorized in (Soviany et al., 2022). It introduces the backward transfer (BWT) metric to quantify the extent of catastrophic forgetting across domains. Based on BWT analysis, Li et al. (2025a) proposes a fixed training order—*code → math → scientific QA → creative writing*—with the goal of minimizing forgetting induced by multi-domain learning.

- **Self-paced CL.** Self-paced CL enables the model to adaptively select training samples according to its learning state. In our implementation, we employ Qwen2.5-7B-Instruct to rank samples by winrate from easy to difficult, and train sequentially following this order. This curriculum reduces the risk of being misled by difficult samples in the early stages, thereby improving stability and promoting better generalization.

- **FAMO.** FAMO is a gradient-balancing approach for MTL. It adjusts loss weights to maximize the improvement rate of the task that progresses the slowest, ensuring that all tasks advance at a comparable pace. This balanced optimization strategy suppresses task dominance and guides the model toward solutions that are both fairer across tasks and stronger in overall performance. FAMO approximates weight updates using historical loss values instead of explicitly computing multi-task gradients, reducing per-iteration time and memory complexity to $\mathcal{O}(1)$. This efficiency makes it particularly suitable for large-scale LLM training.

### C.3   REWARD FUNCTIONS

For all domains, we require the model to enclose its reasoning process within `<think></think>` tags. The reward functions for the four domains are as follows.

- **Math.** We adopt a rule-based reward function:

$$r_{\text{math}}(o, a) = \begin{cases} 1.0, & \text{if } o \text{ has a valid format and } \text{verify}_{\text{math}}(o_{\text{ans}}, a) = \text{true}, \\ -0.5, & \text{if } o \text{ has a valid format but } \text{verify}_{\text{math}}(o_{\text{ans}}, a) = \text{false}, \\ -1.0, & \text{if } o \text{ has an invalid format}, \end{cases}$$

where $o_{\text{ans}}$ denotes the predicted answer extracted from structured tags (e.g., `<answer></answer>`) in the model output $o$, and $\text{verify}_{\text{math}}(\cdot, \cdot)$ checks symbolic equivalence between $o_{\text{ans}}$ and the ground-truth answer $a$ via a deterministic parser (e.g., handling equivalent forms of expressions or equations).

- **Code generation.** We adopt a sandbox-based unit test reward:

$$r_{\text{code}}(o, \text{test\_case}) = \begin{cases} 1.0, & \text{if } o \text{ has a valid format and } \text{exec}(o_{\text{ans}}) \models \text{unittest}(o_{\text{ans}}, \text{test\_case}), \\ -0.5, & \text{if } o \text{ has a valid format but } \text{exec}(o_{\text{ans}}) \not\models \text{unittest}(o_{\text{ans}}, \text{test\_case}), \\ -1.0, & \text{if } o \text{ has an invalid format (syntactically invalid)}, \end{cases}$$

where $o_{\text{ans}}$ is the generated code, executed in a sandbox and validated against the unit tests associated with the sample; $\models$ denotes logical satisfaction.

- **Scientific QA.** We employ a 1.5B General-Verifier[2] (Cheng et al., 2025) to assess consistency between the model's output and the ground-truth answer:

$$r_{\text{qa}}(o, a) = \begin{cases} 1.0 - 0.05 \cdot \min(||o_{\text{ans}}| - |a||, 10), & \text{if } o \text{ has a valid format and } o_{\text{ans}} = a, \\ 0, & \text{if } o \text{ has a valid format but } o_{\text{ans}} \neq a, \\ -1.0, & \text{if } o \text{ has an invalid format}, \end{cases}$$

where $o_{\text{ans}}$ is the extracted answer content. Here, "valid format" means the response adheres to QA conventions (e.g., no garbled text, complete sentences).

- **Creative writing.** We adopt an LLM-as-a-Judge strategy, scoring the model's output $o$ against a reference $o_{\text{ref}}$ via pairwise comparison:

$$r_{\text{writing}}(o, o_{\text{ref}}) = \begin{cases} 1.0, & \text{if } o \text{ has a valid format and } o \succ o_{\text{ref}}, \\ 0.25, & \text{if } o \text{ has a valid format and } o \sim o_{\text{ref}}, \\ -0.5, & \text{if } o \text{ has a valid format and } o \prec o_{\text{ref}}, \\ -1.0, & \text{if } o \text{ has an invalid format}, \end{cases}$$

where $o \succ o_{\text{ref}}$ (preferred), $o \sim o_{\text{ref}}$ (tie), and $o \prec o_{\text{ref}}$ (worse) are determined by a fixed evaluator (Qwen2.5-72B-Instruct) serving as the judge.

### C.4 HYPERPARAMTERS

We use a learning rate of $1 \times 10^{-6}$, a prompt batch size of 128, a mini-batch size of 64, a group size of 8, a rollout temperature of 1.0, $\varepsilon_{\text{low}} = 0.2$, $\varepsilon_{\text{high}} = 0.28$, and $\beta = 0.001$ for CGPO and all baselines. All methods are trained for one epoch. For the mixing coefficient $\alpha$, we tune it within the range of 0.5-1.5, and provide an ablation study on $\alpha$ in Section 4.3.

### C.5 EVALUATION

To comprehensively evaluate cross-domain capabilities, we adopt authoritative benchmarks spanning four domains: **Math**, **Coding**, **Scientific QA**, and **Creative Writing**. The evaluation settings are detailed below:

- **Math domain**
  - **MATH500** (Hendrycks et al., 2021): A set of 500 challenging problems sampled from the full MATH dataset, covering seven areas: elementary algebra, algebra, geometry, number theory, combinatorics, probability, and calculus. Problems are presented in open-ended form and require precise solutions. This benchmark is widely adopted for assessing LLMs' mathematical reasoning and problem-solving abilities.

---

[2]`https://huggingface.co/TIGER-Lab/general-verifier`

- **AMC 2023** (MAA, 2023): A set of 50 questions taken from the AMC 12A and 12B (2023) contests, spanning algebra, geometry, number theory, combinatorics, and probability. Multiple-choice options are removed, requiring models to directly output the final answer. This benchmark focuses on higher-order reasoning, problem analysis, and accurate calculation.

- **Coding domain**

  - **HumanEval** (Chen et al., 2021): Consisting of 164 human-written Python programming tasks, ranging from basic algorithms to medium-level function implementations. It evaluates whether models can generate correct and executable code from natural language descriptions.

  - **MBPP** (Austin et al., 2021): A collection of 974 beginner-level Python problems designed to test the ability to synthesize short programs from natural language instructions. It is a standard benchmark for fundamental code generation.

- **Scientific QA domain**

  - **GPQA (diamond split)** (Rein et al., 2023): Graduate-level QA items written and verified by domain experts across physics, chemistry, biology, and earth sciences. The diamond split represents the most difficult and highest-quality subset, specifically constructed to prevent shallow memorization or pattern matching. To ensure consistent evaluation, we reconstruct ordered option lists using randomized indexing.

  - **SuperGPQA** (Team et al., 2025): Comprising 285 interdisciplinary graduate-level reasoning problems, curated to prevent direct solutions via search engines. To reduce computational cost, we use random seed 42 to sample 200 problems, ensuring both representativeness and reliable measurement of deep reasoning ability.

- **Creative Writing domain**

  - **WritingBench** (Wu et al., 2025): A benchmark of 1000 real-world writing tasks spanning 6 domains and 100 sub-themes, covering diverse styles, task types, and difficulty levels. It evaluates generated text on quality, coherence, creativity, and task alignment through a structured scoring framework. For efficiency, we sample 200 requests using random seed 42, and apply the official critic model `WritingBench-Critic-Model-Qwen-7B`[3] for automated scoring, striking a balance between evaluation cost and representativeness.

## D    MORE DISCUSSIONS

### D.1    APPLICABILITY OF CGPO TO MULTI-DOMAIN PRE-TRAINING

Although our experiments focus on the RL post-training stage, the underlying mechanism of CGPO naturally extends to the multi-domain setting of LLM pre-training. Pre-training corpora are inherently heterogeneous, and the aggregation of losses across diverse domains can lead to a complex optimization landscape. Since CGPO is designed to alleviate such difficulty by leveraging curvature-informed interactions induced by sequential updates, the framework is conceptually agnostic to the specific form of the loss and can, in principle, be applied during pre-training without modification.

It is also worth noting that CGPO is developed to address challenges unique to RL for LLMs, many of which are absent in the pre-training stage. As a result, the design space for multi-domain optimization during pre-training is substantially broader. When the entire corpus is available offline, practitioners may employ a wide range of well-established approaches, including data mixture and sampling strategies (Shukor et al., 2025; Gu et al., 2024), continual or staged domain-specific pre-training (Chen et al., 2025a), and retrieval-augmented pre-training (Izacard et al., 2023; Borgeaud et al., 2022). These techniques are not directly applicable in RL4LLMs but can be highly effective during pre-training, making the relative advantage of CGPO in this setting an open empirical question.

---

[3]https://huggingface.co/AQuarterMile/WritingBench-Critic-Model-Qwen-7B

## D.2 Non-Uniform Domain Importance in Multi-Domain Training

In practical multi-domain applications, different domains may carry different levels of importance. While the main paper focuses on the uniform-weight objective

$$\mathcal{J}(\theta) = \frac{1}{K} \sum_{k=1}^{K} \mathcal{J}_k(\theta),$$

this choice is primarily for conceptual clarity and to highlight the core contribution of CGPO—namely, its ability to mitigate cross-domain optimization conflicts and improve multi-domain reasoning performance.

The CGPO framework can be naturally extended to settings in which domains are assigned non-uniform importance. Let each domain $k$ be associated with a user-defined weight $w_k$ satisfying $\sum_{k=1}^{K} w_k = 1$. The training objective can then be written as

$$\mathcal{J}(\theta; \mathbf{w}) = \sum_{k=1}^{K} w_k \mathcal{J}_k(\theta) = \frac{1}{K} \sum_{k=1}^{K} (K w_k) \mathcal{J}_k(\theta). \tag{26}$$

This formulation is equivalent to scaling each domain-specific loss and its corresponding gradient by a factor proportional to its importance. Crucially, no modification to the CGPO algorithm is required: the sequential updates, geometric interactions, and final interpolation behave identically as in the uniform-weight case, with the only difference being the importance-adjusted gradient contributions. This property allows CGPO to seamlessly accommodate prioritized tasks, enabling it to model practical multi-domain scenarios in which some domains or skills must be emphasized more heavily than others.

## D.3 Why Joint Learning Cannot Reproduce Our Cross-Domain Mechanism

In this section, we provide additional analysis comparing joint learning with the proposed sequential mechanism, clarifying why joint learning cannot recover the same cross-domain Hessian–gradient interactions.

### D.3.1 Sequential Updates Induce Clean Cross-Domain Interactions

As shown in Eq. (5) of the main paper, a ***single*** sequential pass over the domains—corresponding to ***one parameter update***—yields, up to $\mathcal{O}(\eta^2)$,

$$\phi_K - \phi_0 = -\frac{\eta}{K} \sum_{k=1}^{K} \mathbf{g}_k(\phi_0) + \frac{\eta^2}{K^2} \sum_{k=1}^{K} \sum_{l=1}^{k-1} \mathbf{H}_{\sigma(k)}(\phi_0) \mathbf{g}_{\sigma(l)}(\phi_0) + \mathcal{O}(\eta^2). \tag{27}$$

where $\sigma$ is the random permutation sampled at this iteration.

Crucially, the expression above describes the update for a fixed permutation $\sigma$. Since our algorithm re-samples $\sigma$ independently at each iteration, the relevant quantity for understanding the behavior of the sequential mechanism is the ***expectation over*** $\sigma$. Taking expectation symmetrizes the pairwise interactions: each ordered pair $(i, j)$ appears with equal probability. After symmetrization, we have $\mathbf{H}_i(\phi_0) \mathbf{g}_j(\phi_0) + \mathbf{H}_j(\phi_0) \mathbf{g}_i(\phi_0) = \frac{\partial}{\partial \phi_0} \left( \mathbf{g}_i(\phi_0)^\top \mathbf{g}_j(\phi_0) \right)$ (see Appendix B.4), yielding an interpretable alignment effect across domains.

### D.3.2 Two-Step Joint Learning Yields Mixed Second-Order Terms

To analyze why joint learning cannot replicate this mechanism, consider two consecutive joint-training updates. Let

$$\mathcal{L}(\theta) = \frac{1}{K} \sum_{k=1}^{K} \mathcal{L}_k(\theta), \quad \mathbf{g}(\theta) = \nabla \mathcal{L}(\theta) = \frac{1}{K} \sum_{k=1}^{K} \mathbf{g}_k(\theta), \quad \mathbf{H}(\theta) = \nabla^2 \mathcal{L}(\theta) = \frac{1}{K} \sum_{k=1}^{K} \mathbf{H}_k(\theta).$$

Performing two gradient-descent steps with step size $\eta$—note that unlike the sequential pass above, these constitute ***two separate parameter updates***—and expanding up to second order gives

$$\theta_{t+2} - \theta_t \approx -2\eta\,\mathbf{g}(\theta_t)\ +\ \eta^2\,\mathbf{H}(\theta_t)\,\mathbf{g}(\theta_t)$$

$$= -\frac{2\eta}{K}\sum_{k=1}^{K}\mathbf{g}_k(\theta_t) + \frac{\eta^2}{K^2}\sum_{1\leq i\neq j\leq K}\mathbf{H}_i(\theta_t)\mathbf{g}_j(\theta_t) + \frac{\eta^2}{K^2}\sum_{k=1}^{K}\mathbf{H}_k(\theta_t)\mathbf{g}_k(\theta_t). \qquad (28)$$

This expression reveals three types of contributions:

1. **Single-domain gradients** $\mathbf{g}_k(\theta_t)$;
2. **Cross-domain Hessian–gradient interactions** $\mathbf{H}_i(\theta_t)\mathbf{g}_j(\theta_t)$ for $i \neq j$;
3. **Self-curvature terms** $\mathbf{H}_k(\theta_t)\mathbf{g}_k(\theta_t)$.

The presence of the self-curvature terms is the key structural difference from Eq. (27). Because both updates in joint learning are taken with respect to the ***same aggregated loss***, these self-curvature components naturally arise and are typically of comparable magnitude to the cross-domain terms. As a result, they can ***partially or fully cancel*** cross-domain contributions depending on curvature structure. Thus, joint learning does not isolate cross-domain interactions. Its second-order structure is an inseparable mixture of self- and cross-terms, lacking the clean symmetry and interpretability obtained under the sequential scheme.

### D.3.3 Implications for Gradient Alignment

Because joint learning yields both $\mathbf{H}_i\mathbf{g}_j$ and $\mathbf{H}_k\mathbf{g}_k$ terms, the effective update cannot be reduced to a symmetric pairwise structure. In particular, it cannot be rewritten as the gradient of an inter-domain alignment quantity such as $\mathbf{g}_i^\top\mathbf{g}_j$. The self-curvature terms disrupt this symmetry, preventing the simplification that underlies the alignment interpretation in our method.

By contrast, our sequential scheme avoids $\mathbf{H}_k\mathbf{g}_k$ entirely: each domain is updated once per sequential pass, and its gradient is evaluated only after perturbations induced by ***other*** domains. Combined with the expectation over random permutations, this yields a clean, symmetric second-order term capturing cross-domain interactions.

### D.4 Why Approximate Variants of Newton's Method Are Infeasible for RL Training of LLMs

Second-order optimization methods broadly aim to exploit curvature information—typically through matrix-based preconditioning—to enable more geometrically informed parameter updates. These approaches span a wide family of techniques, including Kronecker-factorized natural-gradient methods, layer-wise matrix preconditioners, and approximate Newton-style updates. To illustrate why such methods become impractical in RL training of LLMs, we examine three of the most representative and advanced instances in this family—K-FAC (Martens & Grosse, 2015), Shampoo (Gupta et al., 2018), and SOAP (Vyas et al., 2025)—and analyze the computational and memory implications of applying their core mechanisms at LLM scale.

### D.4.1 K-FAC

K-FAC (Martens & Grosse, 2015) is a Kronecker-factored approximation to natural gradient descent. For a fully-connected (or linear) layer with weight matrix $\mathbf{W} \in \mathbb{R}^{d_{\text{out}} \times d_{\text{in}}}$, input activations $\mathbf{a} \in \mathbb{R}^{d_{\text{in}}}$, and backpropagated output gradients $\mathbf{g} \in \mathbb{R}^{d_{\text{out}}}$, the gradient can be written (for a single sample) as $\nabla_{\mathbf{W}}\mathcal{L} = \mathbf{g}\mathbf{a}^\top$. If we vectorize $\mathbf{W}$ into $\mathbf{w} = \text{vec}(\mathbf{W}) \in \mathbb{R}^{d_{\text{out}}d_{\text{in}}}$, the Fisher information block corresponding to $\mathbf{w}$ is

$$\mathbf{F}_{\mathbf{w}} = \mathbb{E}\big[\nabla_{\mathbf{w}}\mathcal{L}\,\nabla_{\mathbf{w}}\mathcal{L}^\top\big].$$

Under the standard K-FAC independence assumptions (approximately independent $\mathbf{a}$ and $\mathbf{g}$ and certain factorization properties), this block is approximated as a Kronecker product

$$\mathbf{F}_{\mathbf{w}} \approx \mathbf{A} \otimes \mathbf{G}, \qquad \mathbf{A} = \mathbb{E}\big[\mathbf{a}\mathbf{a}^\top\big], \quad \mathbf{G} = \mathbb{E}\big[\mathbf{g}\mathbf{g}^\top\big], \qquad (29)$$

where $\mathbf{A} \in \mathbb{R}^{d_{\mathrm{in}} \times d_{\mathrm{in}}}$ and $\mathbf{G} \in \mathbb{R}^{d_{\mathrm{out}} \times d_{\mathrm{out}}}$ are the Kronecker factors maintained as running (exponential moving) averages over mini-batches.

**Preconditioned update.** Natural gradient descent would apply $\mathbf{F}_{\mathbf{w}}^{-1}$ to the gradient $\nabla_{\mathbf{w}} \mathcal{L}$. Using the approximation in Eq. (29) and the Kronecker identity $(\mathbf{A} \otimes \mathbf{G})^{-1} = \mathbf{A}^{-1} \otimes \mathbf{G}^{-1}$, one obtains the K-FAC preconditioned update for the weight matrix:

$$\Delta \mathbf{W} \approx -\eta \cdot \mathbf{G}^{-1} \cdot \nabla_{\mathbf{W}} \mathcal{L} \cdot \mathbf{A}^{-1}, \qquad \mathbf{W}_{t+1} = \mathbf{W}_t + \Delta \mathbf{W}, \tag{30}$$

where $\eta$ is the learning rate. In practice, $\mathbf{A}^{-1}$ and $\mathbf{G}^{-1}$ are not formed explicitly: K-FAC performs eigendecompositions

$$\mathbf{A} = \mathbf{U}_A \mathbf{\Lambda}_A \mathbf{U}_A^\top, \qquad \mathbf{G} = \mathbf{U}_G \mathbf{\Lambda}_G \mathbf{U}_G^\top,$$

and then applies inverse (or inverse square-root) scalings in these eigen-bases. This requires storing the factors $\mathbf{A}, \mathbf{G}$ (and often their eigenvectors $\mathbf{U}_A, \mathbf{U}_G$) and repeatedly computing or reusing their eigendecompositions.

**Memory cost at LLM scale.** Consider a typical transformer block with hidden size $d_{\mathrm{in}} \approx d_{\mathrm{out}} \approx d$. For modern LLMs, $d$ is in the range $[4096, 8192]$. Each Kronecker factor $\mathbf{A}$ or $\mathbf{G}$ is then a dense $d \times d$ matrix. It is important to note that, for numerical stability, K-FAC implementations typically store curvature factors in at least FP32, even when the model itself uses FP16/BF16. A dense $d \times d$ FP32 matrix requires $4d^2$ bytes. For $d = 4096$, $d^2 = 4096^2 = 16{,}777{,}216$ entries, which leads to

$$\text{size of one factor } (\mathbf{A} \text{ or } \mathbf{G}) \approx 16{,}777{,}216 \times 4 \text{ bytes} \approx 64 \text{ MB}.$$

Thus storing both $\mathbf{A}$ and $\mathbf{G}$ for *one* weight matrix consumes about $2 \times 64 \text{ MB} \approx 128 \text{ MB}$. A transformer block at this width typically has multiple large projection matrices, such as self-attention projections $\mathbf{W}_Q, \mathbf{W}_K, \mathbf{W}_V, \mathbf{W}_O$ plus two large feed-forward matrices. Even if we conservatively apply K-FAC only to four matrices per block (e.g., $\mathbf{W}_Q, \mathbf{W}_K, \mathbf{W}_V, \mathbf{W}_O$) and ignore the FFN, the curvature state per block is already

$$\text{curvature per block} \approx 4 \times 128 \text{ MB} = 512 \text{ MB}.$$

For a 7B LLM with roughly $L \approx 80$ transformer blocks, this yields

$$\text{extra K-FAC curvature memory} \approx 512 \text{ MB} \times 80 \approx 40 \text{ GB } \textit{per device},$$

*only* for storing $\mathbf{A}$ and $\mathbf{G}$ in FP32, without caching eigenvectors.

In practice, many K-FAC variants also cache eigendecompositions, i.e., $\mathbf{U}_A, \mathbf{U}_G$ for each factor. Each eigenvector matrix $\mathbf{U}_A$ or $\mathbf{U}_G$ is again a $d \times d$ FP32 matrix (another $\sim 64$MB for $d = 4096$), effectively doubling the curvature state:

$$\text{curvature per weight } (\mathbf{A}, \mathbf{G}, \mathbf{U}_A, \mathbf{U}_G) \approx 4 \times 64 \text{ MB} = 256 \text{ MB},$$
$$\text{curvature per block (4 weights)} \approx 4 \times 256 \text{ MB} = 1 \text{ GB},$$
$$\text{curvature for 80 blocks} \approx 80 \text{ GB } \textit{per device}.$$

Thus, for a realistic configuration (FP32 factors + cached eigen-bases), even a 7B model with $d = 4096$ requires on the order of 40-80GB of *additional* curvature memory *per device*.

This curvature memory is replicated across data-parallel workers: each device maintains its own copy of the K-FAC state and participates in all-reduce operations to aggregate the factors. The cost is therefore *not* amortized across 8 devices; it is incurred independently on each device.

**Interaction with memory budget of 80GB.** On devices with 80GB memory, RL training of LLMs already pushes device memory close to saturation due to:

- model parameters (for a 7B model in FP16, parameters alone occupy $\sim$14-16GB),

- optimizer states (Adam or AdamW typically add at least another $\sim$2-4$\times$ parameter size, though sharding/ZeRO may partially mitigate this),

- activations and KV caches from long-context rollouts (often tens of GB for sequence lengths in the thousands).

Even under optimistic assumptions with aggressive activation checkpointing and optimizer sharding, reserving an extra $40$-$80$GB purely for K-FAC curvature is incompatible with the 80GB memory budget. There is simply no room left for long-context RL rollouts or for scaling to larger models.

Moreover, this overhead *scales quadratically* with the hidden size $d$. If we increase to $d = 8192$ (typical of larger LLMs), then $d^2 = 8192^2 = 67{,}108{,}864$ entries, which leads to

$$\text{size of one FP32 factor} \approx 67{,}108{,}864 \times 4 \text{ bytes} \approx 256 \text{ MB}.$$

Repeating the above estimates, even storing only $\mathbf{A}$ and $\mathbf{G}$ (no eigenvectors) for four matrices per block across $L$ blocks yields

$$\text{extra curvature memory} \ \sim\ \mathcal{O}\big(L \cdot 4 \cdot 2d^2\big) \ \approx\ \text{tens to over 100 GB per device}$$

for realistic depths and widths. Thus, at LLM scales, K-FAC curvature storage alone can easily demand $50$-$100$GB or more per device, making it infeasible on current 80GB accelerators, especially in RL settings where rollout activations are also resident in memory.

**Computation cost.** K-FAC's main computational bottleneck is computing and updating the eigendecompositions of $\mathbf{A}$ and $\mathbf{G}$ for each layer. The complexity of eigendecomposition for a dense $d \times d$ matrix is $\mathcal{O}(d^3)$, and this dominates the cost of forming the inverse (or inverse square-root) factors.

For $d = 4096$, $d^3 = 4096^3 = 68{,}719{,}476{,}736 \approx 6.9 \times 10^{10}$ FLOPs. Each K-FAC update of a single factor (either $\mathbf{A}$ or $\mathbf{G}$) therefore costs on the order of $10^{11}$ floating-point operations when accounting for constant factors. For four large matrices per block and $L \approx 80$ blocks, a full curvature refresh (updating both $\mathbf{A}$ and $\mathbf{G}$ for all K-FAC blocks) involves on the order of

$$\underbrace{(2 \text{ factors}) \times (4 \text{ matrices}) \times 80}_{\text{number of eigendecompositions}} \times 6.9 \times 10^{10} \ \approx\ 4.4 \times 10^{13} \text{ FLOPs}$$

per curvature update.

In classical applications of K-FAC, these expensive updates are amortized by refreshing curvature only every $\tau$ steps (e.g., $\tau \in [50, 200]$) and reusing the same eigendecomposition in between. Even with such amortization, empirical reports on convolutional and recurrent networks show that K-FAC updates make each optimization step *at least* a few times more expensive than a first-order step when curvature is refreshed regularly. At LLM scale, with many more and much wider layers, the $\mathcal{O}(d^3)$ factor makes this overhead more severe.

When we combine:

- the $\mathcal{O}(d^3)$ eigendecompositions required for each K-FAC factor,

- the need to aggregate curvature statistics across data-parallel workers (extra communication),

- the already high per-step cost of LLM RL training (due to long-context rollouts and large models),

a realistic deployment of K-FAC at LLM scale would very plausibly induce a *three- to five-fold slowdown* in effective optimization throughput compared to standard Adam or AdamW, even if curvature is updated only every $\tau$ steps. Such a slowdown, on top of the massive memory overhead outlined above, renders K-FAC effectively infeasible for RL training of modern LLMs.

### D.4.2 SHAMPOO

Shampoo (Gupta et al., 2018) is a second-order preconditioning method that keeps Kronecker-factored curvature statistics for each weight tensor, and then applies matrix inverse $p$-th roots of these statistics to precondition the gradient. We focus on the matrix case, which already captures the scaling issues at LLM widths.

Consider a matrix parameter $\mathbf{W}_t \in \mathbb{R}^{d_{\text{out}} \times d_{\text{in}}}$ and its (per-minibatch) gradient

$$\mathbf{G}_t \triangleq \nabla_{\mathbf{W}} \mathcal{L}_t \in \mathbb{R}^{d_{\text{out}} \times d_{\text{in}}}.$$

Shampoo maintains two symmetric positive semidefinite (PSD) matrices *per weight matrix*,

$$\mathbf{L}_t = \epsilon \mathbf{I}_{d_{\text{out}}} + \sum_{s=1}^{t} \mathbf{G}_s \mathbf{G}_s^{\top} \in \mathbb{R}^{d_{\text{out}} \times d_{\text{out}}}, \tag{31}$$

$$\mathbf{R}_t = \epsilon \mathbf{I}_{d_{\text{in}}} + \sum_{s=1}^{t} \mathbf{G}_s^{\top} \mathbf{G}_s \in \mathbb{R}^{d_{\text{in}} \times d_{\text{in}}}, \tag{32}$$

where $\epsilon > 0$ is a small damping constant. In practice, $\mathbf{L}_t$ and $\mathbf{R}_t$ are updated by rank-$d_{\text{in}}$ and rank-$d_{\text{out}}$ increments of the form $\mathbf{G}_t \mathbf{G}_t^{\top}$ and $\mathbf{G}_t^{\top} \mathbf{G}_t$ on every optimization step.

The Shampoo update preconditions the gradient with inverse $p$-th powers of $\mathbf{L}_t$ and $\mathbf{R}_t$. For a matrix parameter (order-2 tensor), the original analysis leads to $p = 4$:

$$\widetilde{\nabla_{\mathbf{W}} \mathcal{L}}_t = \mathbf{L}_t^{-\frac{1}{4}} \mathbf{G}_t \mathbf{R}_t^{-\frac{1}{4}}, \tag{33}$$

$$\mathbf{W}_{t+1} = \mathbf{W}_t - \eta \widetilde{\nabla_{\mathbf{W}} \mathcal{L}}_t, \tag{34}$$

where $\eta > 0$ is the step size. The fractional powers are implemented via eigendecomposition: if $\mathbf{L}_t = \mathbf{U}_L \mathbf{\Lambda}_L \mathbf{U}_L^{\top}$ with $\mathbf{\Lambda}_L = \text{diag}(\lambda_1, \ldots, \lambda_{d_{\text{out}}})$, then

$$\mathbf{L}_t^{-\frac{1}{4}} = \mathbf{U}_L \mathbf{\Lambda}_L^{-\frac{1}{4}} \mathbf{U}_L^{\top} \quad \text{with} \quad \mathbf{\Lambda}_L^{-\frac{1}{4}} = \text{diag}\left(\lambda_1^{-\frac{1}{4}}, \ldots, \lambda_{d_{\text{out}}}^{-\frac{1}{4}}\right),$$

and similarly for $\mathbf{R}_t^{-\frac{1}{4}}$. For numerical stability, both the preconditioners and their eigendecompositions are typically kept in at least 32-bit floating point precision, even when $\mathbf{W}_t$ and $\mathbf{G}_t$ are stored in FP16/BF16.

**Memory cost at LLM scale.** Assume a transformer block where all large matrices have approximately square shape $d_{\text{in}} \approx d_{\text{out}} \approx d$, with $d \in [4096, 8192]$ typical for 7B-70B models. For each weight matrix $\mathbf{W}$, Shampoo maintains:

- Two curvature accumulators $\mathbf{L}_t, \mathbf{R}_t \in \mathbb{R}^{d \times d}$;

- In most practical implementations, the corresponding inverse fourth roots $\mathbf{L}_t^{-\frac{1}{4}}, \mathbf{R}_t^{-\frac{1}{4}}$ are also stored, to avoid recomputing matrix roots every step.

Thus, per weight matrix we have roughly four dense $d \times d$ matrices in FP32:

$$\#\text{floats per curvature state} \approx 4d^2,$$

$$\text{memory per curvature state} \approx 4d^2 \times 4 \text{ bytes} = 16d^2 \text{ bytes}.$$

For $d = 4096$, we have

$$d^2 = 4096^2 = 16{,}777{,}216 \approx 1.68 \times 10^7,$$

$$16d^2 \approx 2.68 \times 10^8 \text{ bytes} \approx 256 \text{ MB}.$$

So a *single* large weight matrix requires on the order of

$$\text{Shampoo curvature memory per weight} \approx 256 \text{ MB}.$$

A transformer block typically contains six large matrices (e.g., $\mathbf{W}_Q, \mathbf{W}_K, \mathbf{W}_V, \mathbf{W}_O$ and two feed-forward matrices), so per block we obtain

$$\text{curvature memory per block} \approx 6 \times 256 \text{ MB} = 1536 \text{ MB} \approx 1.5 \text{ GB}.$$

For a 7B-scale model with $d = 4096$ and about $N_{\text{block}} = 40$ transformer blocks, the total Shampoo curvature memory on *one device* is

$$\text{curvature memory } \textit{per device} \approx 1.5 \text{ GB} \times N_{\text{block}} \approx 1.5 \text{ GB} \times 40 \approx 60 \text{ GB}. \tag{35}$$

For a larger 13B-scale model with $d \approx 5120$ and the same number of blocks, the $d^2$ scaling yields

$$d = 5120 \implies \text{curvature memory} \approx 90\text{-}100 \text{ GB } \textit{per device},$$

and for even wider 70B-scale models with $d \approx 8192$, the full-matrix Shampoo preconditioners alone would require several hundred GB of memory.

Crucially, these curvature statistics are optimizer state: in a standard data-parallel RL fine-tuning setup without dedicated sharding of optimizer states (such as Distributed Shampoo), each device replica keeps its own copy of $\mathbf{L}_t, \mathbf{R}_t$ and their inverse roots for its local shard of parameters. This memory is *in addition* to:

- Model parameters (often stored in FP16/BF16 together with first/second-moment optimizer states),
- Activations and attention KV caches required both for backpropagation and for collecting long-context trajectories,
- The auxiliary models typically involved in RLHF pipelines (e.g., reward/scoring models and reference policies), even in setups that do not maintain an explicit critic network.

Empirically, even first-order RLHF baselines (Adam/AdamW) already bring a 7B policy close to the 80 GB limit once the policy, reward/scoring model, and reference model are all active, especially with sequence lengths $\geq 1024$ and realistic batch sizes. Back-of-the-envelope estimates and open-source RLHF reports indicate that a 7B RLHF pipeline can easily consume $\sim$ 60-70 GB of memory on each device *without* any second-order optimizer states. Combining this with the $\sim$ 60 GB of additional curvature memory estimated in Eq. (35) would clearly exceed the 80 GB device capacity. In other words, full-matrix Shampoo at LLM scale effectively leaves no headroom for rollouts, auxiliary models, or even storing the policy itself on a single device (80GB).

**Computation cost.** The two main sources of extra compute in Shampoo are:

- Updating curvature accumulators $\mathbf{L}_t$ and $\mathbf{R}_t$;
- Computing matrix inverse 1/4-powers $\mathbf{L}_t^{-\frac{1}{4}}$ and $\mathbf{R}_t^{-\frac{1}{4}}$.

**(1) Curvature updates.** For each weight matrix,

$$\mathbf{L}_t = \mathbf{L}_{t-1} + \mathbf{G}_t\mathbf{G}_t^\top, \qquad \mathbf{R}_t = \mathbf{R}_{t-1} + \mathbf{G}_t^\top\mathbf{G}_t.$$

Forming the products $\mathbf{G}_t\mathbf{G}_t^\top$ and $\mathbf{G}_t^\top\mathbf{G}_t$ costs

$$\mathcal{O}(d_{\text{out}}^2 d_{\text{in}} + d_{\text{in}}^2 d_{\text{out}}) \approx \mathcal{O}(d^3)$$

FLOPs when $d_{\text{in}} \approx d_{\text{out}} \approx d$. For a transformer with $N_{\text{block}}$ blocks and roughly six large matrices per block, the per-step curvature update cost scales as

$$\text{FLOPs}_{\text{curv}} \approx C_{\text{curv}} N_{\text{block}} d^3, \tag{36}$$

for some modest constant $C_{\text{curv}}$ (approximately $\mathcal{O}(10)$ when counting all $\mathbf{G}\mathbf{G}^\top$ and $\mathbf{G}^\top\mathbf{G}$ computations per block).

For $d = 4096$ and $N_{\text{block}} = 80$:

$$d^3 = 4096^3 = 68{,}719{,}476{,}736 \approx 6.87 \times 10^{10},$$

so Eq. (36) gives

$$\text{FLOPs}_{\text{curv}} \sim 10 \times 80 \times 6.9 \times 10^{10} \approx 5\text{--}7 \times 10^{13} \text{ FLOPs per optimization step,}$$

just to update Shampoo's second-moment statistics for the large matrices in the network.

**(2) Inverse fourth roots.** Computing $\mathbf{L}_t^{-\frac{1}{4}}$ and $\mathbf{R}_t^{-\frac{1}{4}}$ requires either:

- Eigendecomposition or SVD (apply $-1/4$ to eigenvalues), or
- Iterative inverse-square-root schemes (e.g., Newton–Schulz),

both of which cost $\mathcal{O}(d^3)$ per factor. One transformer block with six large matrices has twelve such factors ($\mathbf{L}$ and $\mathbf{R}$ for each weight), giving

$$\text{FLOPs}_{\text{roots, per update}} \approx C_{\text{root}} \times 12 \times d^3,$$

where $C_{\text{root}}$ depends on solver details.

Root updates are typically amortized by refreshing them every $\tau$ optimizer steps. With $N_{\text{block}}$ blocks,

$$\text{FLOPs}_{\text{roots, per step}} \approx \frac{C_{\text{root}} \times 12 \times N_{\text{block}} \times d^3}{\tau}. \tag{37}$$

For $d = 4096$, $N_{\text{block}} = 80$, and $\tau = 100$:

$$\text{FLOPs}_{\text{roots, per step}} \approx \frac{12 \times 80 \times 6.9 \times 10^{10}}{100} \approx 3 \times 10^{11} \text{ FLOPs per step.}$$

The total extra work per optimization step is therefore

$$\text{FLOPs}_{\text{Shampoo extra}} \approx \text{FLOPs}_{\text{curv}} + \text{FLOPs}_{\text{roots, per step}}$$
$$\approx 5 \times 10^{13} + 3 \times 10^{11} \approx \mathcal{O}(10^{13}) \text{ FLOPs per step.} \tag{38}$$

**Relative slowdown from FLOPs.** Large-scale RLHF pipelines for LLMs already require substantial per-step compute due to multiple forward/backward passes (policy, reference, reward/scoring, etc.) and long-context sequences. Eq. (38) shows that full-matrix Shampoo introduces an additional $\mathcal{O}(10^{13})$ FLOPs *per optimization step*, which is typically comparable to—or larger than—the cost of the remainder of the RL update.

Thus, even without invoking any specific algorithmic details, full-matrix Shampoo is expected to induce a multi-$\times$ reduction in optimization throughput solely from its second-order computations.

When combined with the $\sim 60$ GB curvature memory from Eq. (35), the method becomes impractical for RL training of LLMs on 80GB systems:

- The curvature state alone exceeds the available memory once policy, reference, and reward models are included;
- The extra $\mathcal{O}(10^{13})$ FLOPs per step impose a several-fold slowdown relative to standard first-order optimizers.

In short, full-matrix Shampoo cannot be used for RL training of modern LLMs on currently available hardware.

### D.4.3 SOAP

SOAP (Vyas et al., 2025) is a second-order optimizer built on top of Shampoo. For a fully-connected (or linear) layer with weight matrix $\mathbf{W} \in \mathbb{R}^{d_{\text{out}} \times d_{\text{in}}}$ and gradient

$$\mathbf{G} \triangleq \nabla_{\mathbf{W}} \mathcal{L} \in \mathbb{R}^{d_{\text{out}} \times d_{\text{in}}},$$

Shampoo maintains two curvature matrices that approximate second-moment information along the output and input dimensions:

$$\mathbf{L}_t = \beta_2 \, \mathbf{L}_{t-1} + (1 - \beta_2) \, \mathbf{G}_t \mathbf{G}_t^\top, \tag{39}$$

$$\mathbf{R}_t = \beta_2 \, \mathbf{R}_{t-1} + (1 - \beta_2) \, \mathbf{G}_t^\top \mathbf{G}_t, \tag{40}$$

where $\mathbf{L}_t \in \mathbb{R}^{d_{\text{out}} \times d_{\text{out}}}$ and $\mathbf{R}_t \in \mathbb{R}^{d_{\text{in}} \times d_{\text{in}}}$ are updated as exponential moving averages, and $\beta_2 \in (0, 1)$ is a decay coefficient.

**Preconditioned update in the eigenbasis.** SOAP periodically (every $\tau$ steps) computes eigendecompositions of the Shampoo preconditioners:

$$\mathbf{L}_t = \mathbf{Q}_L \, \mathbf{\Lambda}_L \, \mathbf{Q}_L^\top, \qquad \mathbf{Q}_L \in \mathbb{R}^{d_{\text{out}} \times d_{\text{out}}}, \tag{41}$$

$$\mathbf{R}_t = \mathbf{Q}_R \, \mathbf{\Lambda}_R \, \mathbf{Q}_R^\top, \qquad \mathbf{Q}_R \in \mathbb{R}^{d_{\text{in}} \times d_{\text{in}}}, \tag{42}$$

where $\mathbf{\Lambda}_L$ and $\mathbf{\Lambda}_R$ are diagonal matrices of eigenvalues, and $\mathbf{Q}_L$, $\mathbf{Q}_R$ collect the corresponding eigenvectors. SOAP then rotates the gradient into this slowly changing eigenbasis:

$$\mathbf{G}'_t = \mathbf{Q}_L^\top \, \mathbf{G}_t \, \mathbf{Q}_R, \tag{43}$$

and runs Adam-style first- and second-moment updates in the rotated coordinates:

$$\mathbf{M}'_t = \beta_1 \mathbf{M}'_{t-1} + (1 - \beta_1) \mathbf{G}'_t, \tag{44}$$

$$\mathbf{V}'_t = \beta'_2 \mathbf{V}'_{t-1} + (1 - \beta'_2) (\mathbf{G}'_t \odot \mathbf{G}'_t), \tag{45}$$

where $\mathbf{M}'_t, \mathbf{V}'_t \in \mathbb{R}^{d_{\text{out}} \times d_{\text{in}}}$, $\beta_1, \beta'_2 \in (0, 1)$ are Adam-style coefficients, and $\odot$ denotes element-wise multiplication. The preconditioned update in the eigenbasis is

$$\mathbf{U}'_t = \mathbf{M}'_t \oslash (\sqrt{\mathbf{V}'_t} + \varepsilon), \tag{46}$$

where $\oslash$ is element-wise division and $\varepsilon > 0$ is a small numerical constant. Finally, SOAP rotates this update back to the original parameter space:

$$\Delta \mathbf{W}_t = -\eta \mathbf{Q}_L \mathbf{U}'_t \mathbf{Q}_R^\top, \tag{47}$$

$$\mathbf{W}_{t+1} = \mathbf{W}_t + \Delta \mathbf{W}_t, \tag{48}$$

where $\eta > 0$ is the learning rate. Thus SOAP can be viewed as running Adam on a rotated version of the gradient, where the rotation is given by the Shampoo preconditioner eigenbasis.

**Optimizer state and memory cost at LLM scale.** Consider a transformer block where $d_{\text{out}} \approx d_{\text{in}} \approx d$ and the weight matrices are of size $d \times d$. At SOAP scale, the optimizer state associated with a single such matrix $\mathbf{W}$ includes:

- Shampoo curvature matrices $\mathbf{L}_t, \mathbf{R}_t$ (each $d \times d$),
- eigenvector matrices $\mathbf{Q}_L, \mathbf{Q}_R$ (each $d \times d$),
- rotated Adam moments $\mathbf{M}'_t, \mathbf{V}'_t$ (each $d \times d$).

Altogether, this is six dense $d \times d$ matrices per weight matrix.

For numerical stability, these matrices are typically stored in at least FP32, even when the model weights and activations are in BF16/FP16. A single dense $d \times d$ FP32 matrix requires $4d^2$ bytes. Therefore, the SOAP-related optimizer state per weight matrix is

$$\text{bytes per weight (SOAP state)} = 6 \times 4d^2 = 24d^2 \text{ bytes.} \tag{49}$$

Let us instantiate this for a modern LLM width of $d = 4096$:

$$d^2 = 4096^2 = 16{,}777{,}216,$$

$$24d^2 = 24 \times 16{,}777{,}216 = 402{,}653{,}184 \text{ bytes.}$$

Dividing by $1024^2$ to convert to MiB, we have

$$\text{SOAP state per weight} \approx \frac{402{,}653{,}184}{1024^2} \approx 384 \text{ MiB.}$$

Thus, each *single* $4096 \times 4096$ weight matrix carries roughly $384$ MiB of SOAP-specific state.

A typical transformer block at this width has at least four large projection matrices (for self-attention: $\mathbf{W}_Q, \mathbf{W}_K, \mathbf{W}_V, \mathbf{W}_O$), not counting the feed-forward network. Even if we conservatively apply SOAP only to these four matrices, the curvature and moment state per block is

$$\text{SOAP state per block} \approx 4 \times 384 \text{ MiB} = 1536 \text{ MiB} \approx 1.5 \text{ GB.} \tag{50}$$

For a 7B-parameter LLM with roughly $L \approx 80$ transformer blocks, we obtain

$$\text{total SOAP state} \approx 1.5 \text{ GB} \times 80 = 120 \text{ GB } \textit{per device}, \tag{51}$$

*only* counting the FP32 matrices listed above, and ignoring any additional buffers or implementation overhead.

Crucially, this optimizer state is ***replicated*** across data-parallel devices: each worker maintains its own copy of $\mathbf{L}_t, \mathbf{R}_t, \mathbf{Q}_L, \mathbf{Q}_R, \mathbf{M}'_t, \mathbf{V}'_t$ for its local parameters, and participates in all-reduce operations for gradient aggregation. The 120 GB figure in Eq. (51) is therefore a ***per-device requirement***; it is not amortized across multiple devices.

**Interaction with the memory budget in RL training.** On devices with 80GB, RL training of LLMs already pushes device memory close to saturation due to:

- model parameters (for a 7B model in FP16, parameters alone occupy $\sim$14-16GB),
- optimizer states (Adam or AdamW typically add at least another $\sim$2-4$\times$ parameter size, though sharding/ZeRO may partially mitigate this),
- activations and KV caches from long-context rollouts (often tens of GB for sequence lengths in the thousands).

Even under optimistic assumptions with aggressive activation checkpointing and optimizer sharding, it is common to consume on the order of $60$–$70$ GB out of the $80$ GB budget.

Adding the SOAP state from Eq. (51) would require around $120$ GB *per device* purely for curvature and moment information, i.e.,

$$\underbrace{60\text{–}70\text{ GB}}_{\text{existing RL pipeline}} + \underbrace{120\text{ GB}}_{\text{SOAP state}} \gtrsim 180 \text{ GB per device.}$$

This exceeds the $80$ GB memory capacity by more than a factor of two, even before accounting for safety margins and additional framework overhead. In practice, there is simply no configuration (batch size, sequence length, or number of rollout trajectories) that allows both realistic RL training of a 7B LLM and full SOAP optimizer state to coexist on an $80$ GB device.

Moreover, the SOAP memory overhead scales quadratically with the hidden size $d$. If we increase to $d = 8192$ (typical for larger LLMs), then

$$d^2 = 8192^2 = 67{,}108{,}864,$$

$$24d^2 = 24 \times 67{,}108{,}864 = 1{,}610{,}612{,}736 \text{ bytes} \approx 1536 \text{ MiB.}$$

Thus, *one* $8192 \times 8192$ weight would carry about $1.5$ GB of SOAP state, and four such matrices per block over many blocks would push the per-device optimizer state well beyond $200$ GB. Therefore, at realistic LLM widths and depths, the SOAP memory requirements are incompatible with the fixed $80$ GB budget in RL settings.

**Computation cost and slowdown in RL.** SOAP inherits two major computational overheads:

- periodic eigendecompositions of $\mathbf{L}_t$ and $\mathbf{R}_t$ (every $\tau$ steps), and
- per-step rotations of gradients and updates into and out of the preconditioner eigenbasis.

The eigendecomposition of a dense $d \times d$ matrix has complexity $\mathcal{O}(d^3)$. For $d = 4096$,

$$d^3 = 4096^3 = 68{,}719{,}476{,}736 \approx 6.9 \times 10^{10} \text{ FLOPs.}$$

Each SOAP curvature refresh requires two such eigendecompositions per weight (for $\mathbf{L}_t$ and $\mathbf{R}_t$), so the cost per weight matrix is on the order of

$$\text{FLOPs per weight (eigs)} \approx 2 \times 6.9 \times 10^{10} \approx 1.4 \times 10^{11}.$$

With four large matrices per block and $L \approx 80$ blocks, a full curvature refresh involves

$$\text{FLOPs per SOAP refresh} \approx (4 \text{ matrices}) \times (80 \text{ blocks}) \times 1.4 \times 10^{11}$$

$$\approx 4.5 \times 10^{13} \text{ FLOPs.} \tag{52}$$

Even if this cost is amortized by updating the eigenbasis only every $\tau = 100$ steps, the amortized overhead is on the order of $4.5 \times 10^{11}$ FLOPs *per training step*, comparable to or exceeding the cost of the forward-backward pass itself for a 7B model at moderate sequence lengths.

In addition, at every step (not just every $\tau$ steps), SOAP performs the rotations

$$\mathbf{G}'_t = \mathbf{Q}_L^\top \mathbf{G}_t \mathbf{Q}_R, \tag{53}$$

$$\mathbf{U}_t = \mathbf{Q}_L \mathbf{U}'_t \mathbf{Q}_R^\top, \tag{54}$$

which each involve two dense $d \times d$ matrix multiplications (left and right multiplication) and therefore have complexity $\mathcal{O}(d^3)$ per large weight matrix. For $d = 4096$, these rotations add another substantial multiple of $6.9 \times 10^{10}$ FLOPs per weight per step.

When combined across all large matrices and blocks, these extra $\mathcal{O}(d^3)$ operations typically make each SOAP step several times more expensive than a standard Adam/AdamW step. At LLM scale, and especially in RL-style fine-tuning where:

Table 5: Wall-clock time comparison between joint learning and CGPO on extremely large LLMs. Each device has 140GB.

| Model | #Device | Method | 50 steps (hours) | Per step (min) |
|-------|---------|--------|------------------|----------------|
| Qwen2.5-32B-Instruct | 16 | Joint Learning | 7.1 | 8.52 |
|  | 16 | CGPO | 7.5 | 9.00 |
| Qwen2.5-72B-Instruct | 32 | Joint Learning | 12.6 | 15.12 |
|  | 32 | CGPO | 12.8 | 15.36 |

Table 6: **Performance of models (Qwen2.5-7B-Instruct) trained on the multi-domain dataset (math + code) with different methods, evaluated on multiple benchmarks.** The bold font indicates the best result.

| Methods | Math | | Code Generation | | AVG |
|---------|------|-----|-----------------|------|-----|
|  | MATH500 | AMC | HumanEval | MBPP |  |
| FAMO | **76.25** | 57.37 | 84.01 | 71.40 | 72.26 |
| CGPO | 76.15 | **60.81** | **84.66** | **72.60** | ★*73.56* |

- rollouts require long sequences and sufficiently large batch sizes for stable training,

- multiple model passes (policy, reference, reward/scoring, etc.) are performed per update,

- environment interaction and cross-device communication already contribute substantially to the per-step cost,

this optimizer overhead becomes a dominant bottleneck. A conservative estimate is that SOAP would induce at least a **3-5**× slowdown relative to AdamW; for long-context RL training of large LLMs with many wide layers, the combined effect of repeated eigendecompositions and per-step rotations can easily push this into the **5-10**× range in terms of effective tokens-per-second throughput.

Therefore, although SOAP is an attractive optimizer at moderate scales, its quadratic memory footprint and cubic-time eigen-computation render it infeasible for RL training of modern LLMs on current hardwares.

# E    MORE EXPERIMENTS

## E.1    TIMING EXPERIMENTS ON 32B AND 72B MODELS

Extremely large LLMs place substantial computational demand on rollout generation, since the cost of producing each token grows with model size. As model scale increases, rollout generation becomes the dominant component of end-to-end training time, while variations in gradient-update scheduling (e.g., sequential updates vs. a single aggregated update) account for only a small fraction of the total compute.

To quantify this effect, we conduct timing experiments on two large models, Qwen2.5-32B-Instruct and Qwen2.5-72B-Instruct. The 32B and 72B experiments are run on clusters of 16 and 32 devices (140GB per device), respectively. For both models, we measure the total wall-clock time and average per-step time over the first 50 steps. These results provide a representative comparison of computational overhead under realistic large-scale RL training conditions.

As shown in Table 5, across both model scales, the difference between joint learning and CGPO remains marginal relative to the overall training time. This supports the observation that, at extremely large scales, rollout generation dominates end-to-end runtime, and the additional gradient steps used in CGPO do not introduce a meaningful computational bottleneck.

## E.2    DISCUSSION ON THE SENSITIVITY TO THE NUMBER OF DOMAINS

In this section, we provide additional analysis and experiments regarding how CGPO behaves as the number of $K$ increases. As discussed in the main paper, CGPO's effectiveness is driven primarily by

Table 7: **Performance of models (Qwen2.5-7B-Instruct) trained on the multi-domain dataset (math + creative writing) with different methods, evaluated on multiple benchmarks.** The bold font indicates the best result.

| Methods | Math | | Creative Writing | AVG |
|---|---|---|---|---|
| | MATH500 | AMC | WritingBench | |
| FAMO | 74.85 | 54.72 | 64.35 | *64.64* |
| **CGPO** | **75.10** | **58.94** | **67.01** | ★*67.02* |

Table 8: **Performance of models (Qwen2.5-7B-Instruct) trained on the multi-domain dataset (math + code + scientific QA + creative writing + logic + tabular) with different methods, evaluated on multiple benchmarks.** The bold font indicates the best result.

| Methods | Math | | Code Generation | | Scientific QA | | Creative Writing | Logic | Tabular | AVG |
|---|---|---|---|---|---|---|---|---|---|---|
| | MATH500 | AMC | HumanEval | MBPP | GPQA-diamond | SuperGPQA | WritingBench | Zebra | HiTab | |
| FAMO | **75.30** | 55.02 | 82.93 | 68.60 | 22.64 | 31.58 | 63.09 | 36.84 | 68.71 | *56.08* |
| CGPO | 74.90 | **59.84** | **83.88** | **70.80** | **26.91** | **31.72** | **65.08** | **37.63** | **69.57** | ★*57.81* |

the degree of cross-domain conflict, rather than by $K$ itself. Here, we elaborate on this claim and present new experimental evidence.

**Key Observation.** CGPO's sequential curvature-informed mechanism is designed to mitigate cross-domain conflicts. Therefore, its benefit scales with how much the domains disagree. Across all experiments conducted—including those with substantially heterogeneous domain mixtures—we did not observe any evidence of performance plateau or degradation as $K$ increases.

**Experimental Settings.** To empirically verify this, we conducted three groups of experiments, varying either the number of domains or the strength of cross-domain conflict:

1. **Math + Code (moderate conflict):** The datasets and evaluation benchmarks are identical to those used in the main experiments.

2. **Math + Creative Writing (high conflict):** The datasets and evaluation benchmarks are identical to those used in the main experiments.

3. **Math + Code + Scientific QA + Creative Writing + Logic + Tabular:** The datasets and evaluation benchmarks for math, code, scientific QA, and creative writing follow the same setup as in the main experiments. For the logic domain, we train on Zebra Puzzle (1.3k samples) (Lin et al.) and Ordering Puzzle (1.9k samples), and evaluate on the test set of Zebra Puzzle. For the tabular domain, we train on HiTab (4.3k samples) (Cheng et al., 2022) and evaluate on its test set. Both the logic and tabular training and test splits use the filtered versions provided by (Cheng et al., 2025). The reward functions for the logic and tabular domains are rule-based.

We select FAMO for comparison in these experiments because it is the best-performing baseline at the 7B scale in our main experiments. All other training details, reward functions, and evaluation protocols follow the same setup as in the main paper.

**Results.** Across all configurations, CGPO remains stable and effective, as shown in Tables 6-8. Importantly:

- The performance improvement in the math + creative writing setting (high conflict) is noticeably larger than in the math + code setting (moderate conflict), confirming our claim that CGPO's advantage grows as cross-domain conflict increases.

- In the six-domain experiment, CGPO continues to deliver clear, consistent gains, showing that its benefits persist even when $K$ becomes large and the domain mixture is highly heterogeneous.

These results confirm that CGPO's performance does not degrade as the number of domains increases. Instead, its effectiveness is governed by the level of cross-domain conflict, and CGPO remains robust even in large, diverse multi-domain training scenarios.

# F    LIMITATIONS

Although CGPO demonstrates consistent performance improvements across multiple domains, several broader limitations remain. First, similar to existing multi-domain RL4LLMs approaches (Li et al., 2025a), we employ external reward models for certain domains, which may themselves be constrained by current LLM-based evaluation paradigms. For instance, in the creative writing domain, using a single LLM-as-a-judge may introduce stylistic biases that reflect the limits of automated evaluation. Second, similar to existing studies (Cheng et al., 2025; Li et al., 2025b), the overall effectiveness depends on the coverage and granularity of domain-specific rewards, and future advances in reward modeling may naturally enhance performance. Finally, while the randomized sequential update scheme encourages cross-domain interaction, exploring more elaborate scheduling strategies or structured coordination mechanisms remains an open direction for future work. We view these limitations as reflecting broader challenges shared across current RL4LLMs research, and we hope that our work can contribute to the community's continued progress on addressing them.

