# OpenReview forum: "Boosting Multi-Domain Reasoning of LLMs via Curvature-Guided Policy Optimization"
_ICLR.cc/2026/Conference — ICLR 2026 Poster_

### Official Review · Reviewer_SFav · 2025-10-26

**Soundness:** 3
**Presentation:** 4
**Contribution:** 3
**Rating:** 6
**Confidence:** 3

**Summary:**

This manuscript introduces a multi-domain post-training algorithm for LLMs. This method is motivated by Newton's method, where the Hessian matrix can reshape the objective landscape and accelerate convergence. In practice, the authors propose that sequentially training on each domain in a randomized order will naturally simulate the cross-domain Hessian-gradient interaction, and thus is better than joint training on all the domains.

**Strengths:**

* The algorithm is simple in practice and grounded in well-motivated theory.
* The paper is well written; the authors clarify their ideas with a thorough background and intuitive discussion.

**Weaknesses:**

* The author has some concerns about Equation (5) and the discussion following it. The authors claim that "since $\sigma$ is randomized, in expectation every pair will contribute equally". However, for each **specific** training iteration, the permutation is fixed, which means half of the $(i,j)$ pairs will not be present. Therefore, it is not an unbiased estimation of the Hessian matrix, and it is not clear that this effect will cancel out across multiple iterations. The reviewer would like to hear more analysis and discussions on this point.
* Regardless of the above point, let's accept the proposal that the training mechanism in the manuscript can simulate the cross-domain Hessian-gradient interaction. However, it is still unclear why joint-training on all domains cannot. Of course, there are no such interactions for a mini-batch, but there might be some cross-domain interaction stemming from two consecutive mini-batches. To convince the readers of the superiority of the training mechanism proposed in this paper, the authors should provide some analysis of the trivial joint-training mechanism in addition to experimental results.

**Questions:**

* Please refer to the weaknesses.
* I wonder whether the proposed mechanism can be utilized for pre-training of LLMs, because typically the pre-training corpus is also multi-domain. The reviewer would love to hear some discussions on this point, and what is even better, some experimental results.

---

> ### Author Response · Authors · 2025-11-20
> **Response to Reviewer SFav---Part 1/4**
>
> Dear Reviewer SFav,
>
> Thank you for your **valuable review** and for your **high recognition of the practical simplicity and well-motivated theoretical foundation of our algorithm, as well as the clarity and intuitiveness of our overall presentation**. Your positive feedback is truly encouraging and means a great deal to us. Below, we provide detailed responses to your insightful and constructive comments. We have incorporated the corresponding discussions into the appendix and added the necessary references in the main text. **All changes are highlighted in blue.**
>
> Throughout the discussion phase, we genuinely look forward to your feedback and are fully committed to addressing any remaining concerns. If our responses have properly resolved your questions, **we would be deeply grateful if you would consider raising your score**. If anything remains unclear or unconvincing, **we would truly appreciate your further guidance**, and we stand ready to engage promptly and constructively.
>
> ---
>
> > W1: The reviewer has some concerns about Equation (5) and the discussion following it. The authors claim that "since $\sigma$ is randomized, in expectation every pair will contribute equally". However, for each specific training iteration, the permutation is fixed, which means half of the $(i,j)$ pairs will not be present. Therefore, it is not an unbiased estimation of the Hessian matrix, and it is not clear that this effect will cancel out across multiple iterations. The reviewer would like to hear more analysis and discussions on this point.
>
> Thank you for the thoughtful question.
>
> **TL;DR:** Our claim about "every pair contributing equally" is taken in expectation **over the random permutation $\sigma$**, not for any single training step. While a single sampled permutation includes only a subset of ordered pairs, the permutation is redrawn independently a every iteration. Since for any pair $(i,j)$, the probability that $i$ precedes $j$ is exactly one half under a uniform random permutation, the long-run expected contribution of all ordered pairs is symmetric. Thus, the per-step bias introduced by a specific ordering does not accumulate across iterations, and the expansion in Eq. (5) captures the correct expected update direction relevant for our algorithmic interpretation. We have accordingly revised the paper (**Lines 276-281 in Section 3.3**) to make this expectation explicit and avoid the misunderstanding that we are claiming per-step unbiasedness.
>
> **Detailed Response:**
>
> The reviewer is absolutely right that within a single iteration the permutation $\sigma$ is fixed, and consequently only the ordered pairs $(\sigma(l), \sigma(k))$ with $l < k$ appear in Eq. (5). Our analysis does not claim that a single iteration provides an unbiased or complete enumeration of all cross-domain Hessian-gradient terms. Instead, the statement "since $\sigma$ is randomized, in expectation every pair contributes equally" is taken in **expectation over the randomness of $\sigma$**.
>
> Because $\sigma$ is resampled independently at every training step, for any fixed pair of domains $(i,j)$ we have
>
> - ${\rm Pr}[\sigma(i) < \sigma(j)] = 1/2$.
>
> - ${\rm Pr}[\sigma(i) > \sigma(j)] = 1/2$.
>
> Thus across iterations, the contributions $\mathbf{H}_i \mathbf{g}_j$ and $\mathbf{H}_j \mathbf{g}_i$ occur with equal probability and equal weight in expectation. This establishes that the *expected* second-order term is symmetric across all domain pairs, which is precisely the quantity used in the analytic interpretation following Eq. (5). We do not rely on any single iteration being unbiased.
>
> Moreover, because each iteration draws a fresh permutation, the directionally "biased" contributions of any particular ordering do not accumulate systematically; their signs and ordering change from step to step, and only the symmetric expected term remains persistent. This is why the expectation-level analysis correctly reflects the mechanism by which sequential updates induce gradient-alignment effects across domains.

---

> ### Author Response · Authors · 2025-11-20
> **Response to Reviewer SFav---Part 2/4**
>
> > W2: Regardless of the above point, let's accept the proposal that the training mechanism in the manuscript can simulate the cross-domain Hessian-gradient interaction. However, it is still unclear why joint-training on all domains cannot. Of course, there are no such interactions for a mini-batch, but there might be some cross-domain interaction stemming from two consecutive mini-batches. To convince the readers of the superiority of the training mechanism proposed in this paper, the authors should provide some analysis of the trivial joint-training mechanism in addition to experimental results.
>
> Thank you for your insightful question.
>
> **TL;DR:** Although two consecutive joint-training batches may introduce some second-order effects, their structure is fundamentally different from the interactions produced by our sequential mechanism. Specifically, joint training inevitably mixes **self-curvature terms** $\mathbf{H}\_k\mathbf{g}\_k$ with the desired **cross-domain interactions** ($\mathbf{H}\_i\mathbf{g}\_j, i\neq j$). These self-curvature terms can partially or fully cancel the cross-domain contributions, preventing joint training from yielding a clean, symmetric interaction across domains. In contrast, our method, by sequentially updating one domain at a time under a freshly sampled random permutation, isolates the cross-domain Hessian–gradient interactions in expectation and avoids contamination from self-curvature terms. Thus, joint training cannot recover the same second-order structure.  We have added this discussion to **Appendix E.3** and included a reference to it in Section 3.3 (Lines 281-282).
>
> **Detailed Response:**
> The reviewer asks why joint training on all domains, despite lacking second-order interactions within a single mini-batch, might not accumulate comparable cross-domain effects over consecutive batches. To address this, we analyze the two-step behavior of joint training and compare it with the update structure derived for our randomized sequential scheme.
>
> (1) **Our mechanism produces clean cross-domain interactions.** As shown in the main text (Eq. (5)), a single sequential pass over domains---corresponding to **one parameter update**---yields, up to $\mathcal{O}(\eta^2)$:
>
> $$
> \phi\_K - \phi\_0 = -\frac{\eta}{K}\sum\_k \mathbf{g}\_k(\phi\_0) + \frac{\eta^2}{K^2} \sum\_{k > l} \mathbf{H}\_{\sigma(k)}(\phi\_0)\mathbf{g}\_{\sigma(l)}(\phi\_0) + \mathcal{O}(\eta^2).
> $$
>
> The key point---and the source of the theoretical clarity---is that the expansion above is the update for *a fixed permutation* $\sigma$. Because our algorithm re-samples $\sigma$ independently at each iteration, the relevant object for algorithmic behavior is the **expectation** over permutations. Taking expectation symmetrizes the pairwise interactions, ensuring that every ordered pair $(i,j)$ contributes equally; after symmetrization, we have $\mathbf{H}\_i\mathbf{g}\_j + \mathbf{H}\_j\mathbf{g}\_i = \nabla(\mathbf{g}\_i^\top \mathbf{g}\_j)$ (For details please refer to our response to W1). This yields a clean, interpretable cross-domain alignment effect.
>
> (2) **Two-step joint training still does not match our mechanism.** Now consider two consecutive joint-training batches, each containing all domains. Let $\mathcal{L}(\theta)=\frac{1}{K}\sum\_{k=1}^K \mathcal{L}\_k(\theta)$, $\mathbf{g}(\theta) = \nabla \mathcal{L}(\theta) = \frac{1}{K} \sum\_{k=1}^K \mathbf{g}\_k(\theta)$, and $\mathbf{H}(\theta) = \nabla^2 \mathcal{L}(\theta) = \frac{1}{K} \sum\_{k=1}^K \mathbf{H}\_k(\theta)$. Performing two joint steps with step size $\eta$---note that unlike the sequential pass above, these constitue **two separate parameter updates**---and expanding up to second order yields
>
> $$
> \begin{align}
>     \theta\_{t+2} - \theta\_t \approx -2\eta\mathbf{g}(\theta\_t)+\eta^2\mathbf{H}(\theta\_t) \mathbf{g}(\theta\_t) = -\frac{2\eta}{K}\sum\_{k=1}^K \mathbf{g}\_k(\theta\_t) + \frac{\eta^2}{K^2} \sum\_{1\leq i\neq j\leq K} \mathbf{H}\_i(\theta\_t)\mathbf{g}\_j(\theta\_t) + \frac{\eta^2}{K^2} \sum\_{k=1}^K \mathbf{H}\_k(\theta\_t)\mathbf{g}\_k(\theta\_t).
> \end{align}
> $$
>
> which contains three types of terms: (1) single-domain gradients $\mathbf{g}\_k$, (2) cross-domain Hessian–gradient interactions $\mathbf{H}\_i\mathbf{g}\_j$, and (3) self-curvature terms $\mathbf{H}\_k\mathbf{g}\_k$. The crucial difference from our method is the presence---and magnitude---of **self-curvature terms**, which arise naturally because both steps use the full multi-domain loss. These self-curvature terms can be of the same order as the cross-domain terms and can **partially or fully cancel** them, depending on domain curvature structure. Consequently:
>
> - joint training does not isolate cross-domain interactions,
> - the effective second-order structure is a mixture of self- and cross-terms,
> - the clean symmetry exploited in our algorithm is lost.
>
> Thus, even over two steps, joint training does not recover the interaction in Eq. (5).
>
> ***(To be continued..)***

---

> ### Author Response · Authors · 2025-11-20
> **Response to Reviewer SFav---Part 3/4**
>
> ***(Continued from "Response to Reviewer SFav---Part 2/4")***
>
> (3) **Why this prevents joint training from matching our alignment effect.** Because joint training includes both $\mathbf{H}\_i\mathbf{g}\_j$ and $\mathbf{H}\_k\mathbf{g}\_k$, its effective second-order term cannot be written as a clean symmetric pairwise interaction. The contaminating self-curvature components disrupt the alignment mechanism derived in our method, and the interaction does not simplify to the gradient-alignment form shown in the paper.
> In contrast, **our sequential scheme explicitly avoids $\mathbf{H}\_k\mathbf{g}\_k$ terms** because each domain is updated only once per sequential pass and only responds to perturbations induced by *other* domains. Combined with the expectation over random $\sigma$, this yields a clean, interpretable cross-domain effect.

---

> ### Author Response · Authors · 2025-11-20
> **Response to Reviewer SFav---Part 4/4**
>
> > Q1: I wonder whether the proposed mechanism can be utilized for pre-training of LLMs, because typically the pre-training corpus is also multi-domain. The reviewer would love to hear some discussions on this point, and what is even better, some experimental results.
>
> Thank you for the thoughtful question.
>
> **TL;DR:** The core mechanism of CGPO can indeed be naturally applied to the multi-domain setting of LLM pre-training. However, we currently do not know how it would compare against the rich and well-established ecosystem of multi-domain pre-training strategies. We have added this discussion to **Appendix E.1** and included a reference to it in Section 1 (Lines 66-67).
>
> **Detailed Response:**
>
> - We agree that pre-training corpora are inherently multi-domain, and the optimization challenge arising from aggregating heterogeneous domain losses aligns well with the motivation of CGPO. Conceptually, **CGPO addresses the optimization difficulty caused by the complex loss landscape induced by multi-domain aggregation**, and this mechanism is agnostic to the form of the loss. Therefore, **extending CGPO to the pre-training stage is straightforward in principle**.
>
> - **CGPO is designed to address the challenges specific to RL4LLMs**, many of which are *not* bottlenecks during pre-training. This means that many techniques that are infeasible in RL4LLMs become applicable in the pre-training setting, giving practitioners **a much broader range of methodological choices**. For example, in RL4LLMs, responses are generated online, making it impossible to know in advance the importance of a sample or its potential conflict with other samples. In contrast, during pre-training, the entire corpus is available offline, enabling the use of methods such as data mixture and sampling strategies [1,2], continual or staged domain pre-training [3], and retrieval-augmented pre-training [4,5] to enhance multi-domain learning.
>
> - Due to limited computational resources, we are unfortunately unable to conduct full-scale pre-training experiments with CGPO (this is also one reason why our work focuses on post-training). We sincerely appreciate your understanding. Nevertheless, we are genuinely excited about the possibility of applying CGPO to multi-domain pre-training in future work, and we believe it is a promising direction for the community.
>
> **References:**
>
> [1] Scaling Laws for Optimal Data Mixtures.
>
> [2] CMR Scaling Law: Predicting Critical Mixture Ratios for Continual Pre-training of Language Models.
>
> [3] Towards Effective and Efficient Continual Pre-training of Large Language Models.
>
> [4] Atlas: Few-shot Learning with Retrieval Augmented Language Models.
>
> [5] Improving Language Models by Retrieving from Trillions of Tokens.

---

> ### Comment · Reviewer_SFav · 2025-11-20
>
> Thank the authors for the response. Most of my concerns are addressed.
>
> Regarding W1: The reviewer understands that intuitively, the bias on $\theta$ might not accumulate since we sample a new permutation for each iteration. However, this is not trivial to prove, because $\theta_i$ is not independent of $\theta_{i-1}$. In fact, suppose $X_1 \sim p_1(\cdot|\alpha_1), X_2 \sim p_2(\cdot|\alpha_2, X_1)$, it is in general not true that $E_{\alpha_1, \alpha_2} X_2 = E_{\alpha_2} [X_2| X_1=E_{\alpha_1}X_1\]$. The right side of the equation corresponds to the unbiased value, but the pipeline proposed by the authors simulates the left side of the function. I would like to hear more analysis from the authors on this point.

---

> > ### Author Response · Authors · 2025-11-21
> > **Thank you for your prompt and thoughtful reply---Part 1/2**
> >
> > Dear Reviewer SFav,
> >
> > Thank you very much for your prompt and thoughtful reply. We want to begin by saying that **we genuinely enjoy our high-quality and in-depth exchange with you**. Your questions and feedback have been immensely valuable in helping us refine and strengthen our work.
> >
> > We would like to mention that there may have been some **misalignment in understanding** regarding Eq. (5). It seems possible that the way we presented Eq. (5) in the paper may not have fully conveyed its intended meaning, and consequently, our earlier response may not have addressed your concern in the way you intended. To prevent any further confusion, and to ensure that our discussion remains focused on the content of the paper itself, we would first like to clarify what Eq. (5) is intended to express in our work, and what the subsequent expectation over permutations is meant to convey.
> >
> > # (1) What Eq. (5) expresses
> >
> > Eq. (5) results from a **deterministic Taylor expansion** of one sequential update pass in our algorithm, conditioned on a *fixed permutation $\sigma$*. Its purpose is to make explicit that the resulting parameter change naturally decomposes into:
> >
> > - a first-order term corresponding to aggregated gradients, and
> > - a second-order interaction term involving Hessian-gradient products.
> >
> > Importantly, the Hessian-gradient terms in Eq. (5) do not arise from any expectation or statistical averaging. They are simply the deterministic consequence of executing a single sequential update under a fixed ordering. There is no sense in which we are "estimating the Hessian through randomness". Rather Eq. (5) highlights the **structural presence** of cross-domain second-order interactions induced by sequential updates.
> >
> > # (2) Why we take expectation over $\sigma$
> >
> > When we take the expectation over $\sigma$, our intention is to capture a **symmetry property**.
> >
> > To make this more concrete, imagine that at the same parameter $\theta\_t$, we were able---*hypothetically, since the algorithm does not actually do this*---to sample $M$ independent permutations $\\{\sigma\_t^{(m)}\\}\_{m=1}^M$, each corresponding to an ordering $\tau\_t^{(m)}=\left(\sigma\_t^{(m)}(k)\right)\_{k=1}^K$. In this hypothetical scenario, as $M \to \infty$, the events "$i$ appears before $j$" and "$j$ appears before $i$" would occur with essentially equal frequency for every pair $(i,j)$. This limiting symmetry is exactly what our expectation argument is intended to express, and it is what leads to the symmetric combination $\mathbf{H}\_i \mathbf{g}\_j + \mathbf{H}\_j \mathbf{g}\_i$ in the discussion following Eq. (5).
> >
> > In the actual algorithm, of course, we sample **only one** permutation at each iteration. This introduces sampling **error**---but not **bias in the expectation sense**---because we do not average over multiple permutations.
> >
> > Importantly, this sampling error does not accumulate in a harmful way in practice. A helpful way to view this is through **an analogy with standard SGD**: each stochastic gradient is, in expectation, equal to the true gradient (just as the contributions of $\mathbf{H}\_i\mathbf{g}\_j$ and $\mathbf{H}\_j\mathbf{g}\_i$ are symmetric in expectation), yet in practice we use only one stochastic gradient per step rather than averaging many samples---just as our algorithm samples only one permutation per iteration rather than averaging over many permutations at the same parameter. This practice in SGD does introduce variance and error, but it does not undermine either the effectiveness of SGD or the usefulness of the statement that "the stochastic gradient equals the true gradient in expectation". The same phenomenon appears in our algorithm.
> >
> > Therefore, when we refer to an expectation, we mean the conditional expectation taken at a fixed $\theta\_t$, i.e., conditional on the past history $\mathcal{F}\_{t-1}$---just as the expectation of a stochastic gradient in SGD is interpreted conditional on the current parameter value.
> >
> > > ***P.S.*** The SGD analogy may feel somewhat indirect. If this comparison is confusing or unhelpful, please let us know and we would be happy to offer an alternative explanation.
> >
> > ***(To be continued...)***

---

> > ### Author Response · Authors · 2025-11-21
> > **Thank you for your prompt and thoughtful reply---Part 2/2**
> >
> > ***(Continued from "Thank you for your prompt and thoughtful reply---Part 1/2")***
> >
> > # (3) Addressing your concern about dependency across iterations
> >
> > You have raised an important and subtle point: although $\sigma$ is drawn independently at each iteration, the parameter $\theta\_t$ depends on past permutations. Your example involving $X\_1$ and $X\_2$ highlights the risk of incorrectly exchanging expectations when variables are not independent.
> >
> > We fully agree with this concern. However, our analysis does **not** rely on substituting $\theta\_t$ with its expectation. Instead, we use the fact that: **$\sigma$ is independent of the entire history $\mathcal{F}\_{t-1}$, even though $\theta\_t$ depends on that history.** This implies that the probability of any ordering $(i,j)$ remains symmetric after conditioning on $\theta_t$. Therefore, our expectation is always taken in the form $\mathbb{E}[\cdot \mid \mathcal{F}\_{t-1}]$. In other words, **our use of expectation never requires substituting $\theta\_t$ with its mean**, and does not suffer from the issue in your example.
> >
> > The role of the expectation is solely to show that, given $\theta\_t$, every cross-domain interaction term $\mathbf{H}\_i\mathbf{g}\_j$ appears symmetrically in expectation, as $\sigma$ is fully independent and uniformly random.
> >
> > ---
> >
> > We deeply appreciate your insightful questions, which have helped us significantly improve the clarity and rigor of our presentation. Thank you again for the engaging and constructive discussion. We sincerely appreciate your time and expertise.

---

> > > ### Comment · Reviewer_SFav · 2025-11-21
> > >
> > > Thank the authors for the clarification. Now I understand that the discussion following Eq. (5) emphasizes the presence of cross-domain interaction, and the analogy to SGD convinces me about the plausibility and practicality of the proposed pipeline. It may be helpful to reflect this point in the manuscript to improve clarity for readers.
> > >
> > > I genuinely adore the insightfulness of this work, so I have raised my score to 8.

---

> > > > ### Author Response · Authors · 2025-11-21
> > > > **Thank you for your recognition of our work.**
> > > >
> > > > Dear Reviewer SFav,
> > > >
> > > > Thank you for your recognition of our work. Following your suggestion, we have included this discussion as a remark in **Appendix B.4** of the revised paper.
> > > >
> > > > We would also like to express our gratitude once again for your **constructive suggestions** and **insightful questions**, which have significantly improved the quality of our work. **We greatly enjoy our exchange with you**.
> > > >
> > > > Best regards,
> > > >
> > > > Authors of Submission 19328

---

### Official Review · Reviewer_Fgcd · 2025-10-30

**Soundness:** 3
**Presentation:** 2
**Contribution:** 3
**Rating:** 4
**Confidence:** 5

**Summary:**

1．	This paper presents Curvature-Guided Policy Optimization (CGPO), a novel training framework designed to enhance the multi-domain reasoning capabilities of Large Language Models (LLMs) using reinforcement learning (RL).
2．	The core challenge addressed is the presence of cross-domain conflicts, where improvements in one domain (e.g., coding) can lead to degradation in another (e.g., creative writing).
3．	CGPO is inspired by the geometric principles of Newton's method but avoids its prohibitive computational cost by using a randomized sequential update mechanism. This mechanism implicitly approximates Hessian-gradient interactions across domains, fostering gradient alignment and steering the model towards parameters that perform well across all domains simultaneously.
4．	The method is evaluated on a diverse dataset spanning math, coding, science, and creative writing, and demonstrates superior performance over several strong baselines.

**Strengths:**

1.  The core insight of leveraging the geometric structure of the reward landscape (inspired by Newton's method) to mitigate multi-domain conflicts is both innovative and principled. The motivation is clearly laid out, connecting the limitations of existing first-order and gradient-manipulation methods to the potential benefits of second-order information.
2.  A significant contribution is the design of a method that captures the benefits of curvature information without explicitly computing the Hessian. The proposed mechanism of randomized sequential updates is elegant and computationally lightweight, introducing only negligible overhead compared to standard joint training. This makes CGPO highly practical and scalable for large-scale LLM RL training, a critical consideration in this field.
3.  The paper provides a solid mathematical foundation for the proposed method. The derivations (in the main text and appendix) convincingly show how the sequential update scheme approximates cross-domain Hessian-gradient products and, in expectation, encourages gradient alignment. This moves beyond a purely empirical contribution.
4.  The choice of four distinct domains (math, code, science, creative writing) effectively demonstrates the method's ability to handle varied types of reasoning and reward signals.
The comparison includes a representative set of baselines (joint learning, curriculum learning, gradient balancing), showing that CGPO advances the state-of-the-art.
5．Evaluating on both 3B and 7B parameter models strengthens the claims, showing that the benefits are consistent and potentially scale with model size. Using seven established benchmarks provides a robust assessment of multi-domain capability.
6. The ablation studies on domain order randomization and the mixing coefficient `α` are crucial for validating the design choices. The analysis of computational overhead directly addresses a key potential concern for adoption.

**Weaknesses:**

1.  The role and intuition behind the final interpolation step (`θ_new ← φ_0 + α(φ_K - φ_0)`) could be explained more clearly in the main text. While the ablation study shows its importance, a more intuitive explanation of why this interpolation stabilizes training would be helpful. Is it primarily a form of trust region enforcement or a step-size damping mechanism?
2.  The paper would be strengthened by a more explicit discussion of its limitations. For instance, the creative writing reward is based on a single LLM-as-a-judge (Qwen2.5-72B). The potential biases of this judge and how they might influence the overall multi-domain optimization are not discussed.
3.  While the baselines are well-chosen, the related work section mentions other second-order approximation methods like Shampoo, K-FAC, and SOAP. A discussion on why CGPO is a more suitable path for the RL-for-LLMs setting compared to adapting these existing approximate second-order methods would provide valuable context and further justify the novelty. The argument is implicit but could be made more explicit.

**Questions:**

1.How sensitive is CGPO to the number of domains (K)? Does performance plateau or degrade as K becomes very large?
 2. The method relies on multiple gradient steps per "outer" iteration. While the overhead is shown to be low, are there scenarios (e.g., with extremely large models) where even this could become a bottleneck compared to a single-update method?

---

> ### Author Response · Authors · 2025-11-20
> **Response to Reviewer Fgcd---Part 1/5**
>
> Dear Reviewer Fgcd,
>
> Thank you for your **valuable review** and for your **high recognition of the conceptual contributions, methodological soundness, and overall clarity and quality of our work**. Your positive evaluation is truly encouraging and means a great deal to us. Below, we provide detailed responses to your insightful and constructive comments. We have incorporated the corresponding discussions and experiments into the appendix and added the necessary references in the main text. **All changes are highlighted in blue.**
>
> Throughout the discussion phase, we genuinely look forward to your feedback and are fully committed to addressing any remaining concerns. If our responses have properly resolved your questions, **we would be deeply grateful if you would consider raising your score**. If anything remains unclear or unconvincing, **we would truly appreciate your further guidance**, and we stand ready to engage promptly and constructively.
>
> ---
>
> > W1: The role and intuition behind the final interpolation step ($\theta\_{\rm new} \leftarrow \phi\_0 + \alpha (\phi\_K - \phi\_0)$) could be explained more clearly in the main text. While the ablation study shows its importance, a more intuitive explanation of why this interpolation stabilizes training would be helpful. Is it primarily a form of trust region enforcement or a step-size damping mechanism?
>
> Thank you for the helpful suggestion. Below we provide a clearer explanation of the role and intuition behind the final interpolation step, and we have added the corresponding discussion to **Section 3.3 (Lines 283-293)** in our revised paper.
>
> **TL;DR:** The vector $\phi\_K - \phi\_0$ provides a direction that integrates both single-domain gradients and cross-domain Hessian-gradient interactions, while $\alpha$ controls how far we move along this beneficial direction---large enough to leverage its effect, but not so large as to compromise stability.
>
> **Detailed Response:**
>
> - **Where should we move?** As discussed in Lines 258–281, after the sequential updates $\phi\_0 \to \phi\_1 \to \cdots \to \phi\_K$, the displacement $\phi\_K - \phi\_0$ consists of two components: (1) the aggregation of single-domain gradients, capturing domain-specific learning; (2) cross-domain Hessian–gradient products, capturing geometric interactions across domains. This decomposition follows directly by dividing both sides of Eq. (5) by $\alpha$. Hence, $\phi\_K - \phi\_0$ forms a well-informed, geometry-aware update direction that encourages cross-domain alignment.
>
> - **How far should we move?** This is the natural follow-up question once the update direction is determined. CGPO uses the hyperparameter $\alpha$ to balance being aggressive versus being conservative.
>
>     - $\alpha$ cannot be too large. A large $\alpha$ would move the parameters far outside the locally smooth region where the gradient and curvature reliably predicts descent, which can lead to oscillation or divergence---similar to what happens in standard optimization when the learning rate is too large.
>     - $\alpha$ cannot be too small. If $\alpha$ is too small, the update under-utilizes the valuable geometric information encoded in $\phi\_K -\phi\_0$, causing the method to collapse back to a near-identity update and lose the benefits introduced by the cross-domain Hessian–gradient interactions in CGPO.
>     - In ablation studies (Lines 479-485), we observe that $\alpha=0.9, 1.2, 1.5$ all outperform the strongest baseline, indicating that CGPO is robust to $\alpha$ within a reasonable range, while still requiring it to be neither too aggressive nor too conservative.

---

> ### Author Response · Authors · 2025-11-20
> **Response to Reviewer Fgcd---Part 2/5**
>
> > W2: The paper would be strengthened by a more explicit discussion of its limitations. For instance, the creative writing reward is based on a single LLM-as-a-judge (Qwen2.5-72B). The potential biases of this judge and how they might influence the overall multi-domain optimization are not discussed.
>
> Thank you for raising this important point.
>
> **TL;DR:** We have added an explicit Limitations paragraph to the revised paper (**Section 6**), summarizing several practical considerations relevant to CGPO, including the constraints introduced by LLM-based reward models. Regarding your specific question, we (i) acknowledge that a single LLM-as-a-judge may introduce stylistic biases, (ii) explain that our use of Qwen2.5-72B-Instruct reflects a practical cost–effectiveness trade-off given the multi-domain focus of this work, and (iii) clarify that CGPO is fully decoupled from reward-model design and can seamlessly integrate alternative or richer reward constructions.
>
> **Detailed Response:**
>
> (1) **Added Limitations Paragraph in the revised paper.** Following your suggestion, we have added a Limitations paragraph to the revised version (**Section 6**) to provide a balanced and transparent discussion of this work's limitations. This paragraph highlights several practical considerations that may influence the performance of the proposed framework. For your convenience, we quote the added paragraph below:
>
> *Although CGPO demonstrates consistent performance improvements across multiple domains, several broader limitations remain. First, similar to existing multi-domain RL4LLMs approaches, our method employs external reward models for certain domains, which may themselves be constrained by current LLM-based evaluation paradigms. For instance, in the creative writing domain, using a single LLM-as-a-judge may introduce stylistic biases that reflect the limits of automated evaluation. Second, similar to many RL approaches for LLMs, the overall effectiveness depends on the coverage and granularity of domain-specific rewards, and future advances in reward modeling may naturally enhance performance. Finally, while the randomized sequential update scheme encourages cross-domain interaction, exploring more elaborate scheduling strategies or structured coordination mechanisms remains an open direction for future work. We view these limitations as reflecting broader challenges shared across current RL4LLMs research, and we hope that our framework can contribute to the community’s continued progress on addressing them.*
>
> We respectfully believe that this addition provides a clear and responsible overview of the key practical considerations associated with our approach.
>
> (2) **Addressing the Reviewer's Specific Concern about LLM-as-a-Judge Bias.** We acknowledge the inherent biases that may arise when using a single LLM-as-a-judge, explain the practical reasons for choosing Qwen2.5-72B-Instruct in this work, and clarify that CGPO is fully decoupled from the choice of reward model and can seamlessly accommodate alternative or richer reward constructions.
>
> - **First, we acknowledge the inherent limitations of using a single LLM-as-a-judge.** We appreciate your thoughtful question regarding the use of a single LLM-as-a-judge (Qwen2.5-72B-Instruct) for evaluating creative writing responses. We fully acknowledge that any single reward model (RM) may inherit stylistic preferences from the judge itself, which can introduce potential biases. This is a widely recognized limitation of current LLM-based evaluation paradigms, and several alternative strategies (e.g., multi-judge ensembles, calibrated aggregation schemes, or human-in-the-loop scoring) have been proposed to mitigate these effects.
>
> - **Second, our choice of Qwen2.5-72B-Instruct reflects a practical cost–effectiveness trade-off aligned with the focus of this work.** In our setting, we select Qwen2.5-72B-Instruct based on a cost-effectiveness trade-off driven by the focus of our work. Our primary goal is to improve multi-domain RL performance of LLMs, rather than optimizing for peak performance in the creative writing domain alone. Under this objective, Qwen2.5-72B-Instruct strikes a practical balance: it offers strong empirical judging quality while keeping the reward-evaluation cost manageable, which is crucial since every RL update requires repeated calls to the RM.
>
> - **Third, CGPO is fundamentally decoupled from the choice of reward model.** Finally, we emphasize that CGPO is fully decoupled from the choice of RM and does not rely on any particular property of Qwen2.5-72B-Instruct. The framework remains unchanged if one substitutes a different judge, employs a multi-judge ensemble, or incorporates human preference scores. We now highlight this point in the revised paper and view richer reward construction as a valuable future direction.

---

> ### Author Response · Authors · 2025-11-20
> **Response to Reviewer Fgcd---Part 3/5**
>
> > W3: While the baselines are well-chosen, the related work section mentions other second-order approximation methods like Shampoo, K-FAC, and SOAP. A discussion on why CGPO is a more suitable path for the RL-for-LLMs setting compared to adapting these existing approximate second-order methods would provide valuable context and further justify the novelty. The argument is implicit but could be made more explicit.
>
> Thank you for your valuable comment. Following your suggestion, we have added a detailed discussion in **Appendix E.4** explaining why existing approximate second-order methods are not applicable to RL training of LLMs (thereby naturally clarifying why our CGPO is a more suitable choice), and we have added a corresponding reference in **Section 5 (Related Work)**.
>
> The added material spans nearly **eight pages (Pages 24-31)** and provides in-depth analyses of the algorithms and computational costs of K-FAC, Shampoo, and SOAP. For your convenience, we summarize the main conclusions below.
>
> **Summary (see Appendix E.4 for details):** All three approximate second-order methods (K-FAC, Shampoo, SOAP) suffer from (1) additional and prohibitive resource requirements at LLM scale, adding approximately **40-80GB extra memory per GPU** (a single A100 typically has only 80GB), (2) substantial computational cost, typically causing at least a **three-fold slowdown** and in some cases reaching **5-10 times slower training throughput**, and (3) structural mismatch with RL training of LLMs, including long-context rollouts and high-variance gradients.
>
> Therefore, although these methods are attractive at moderate scale, they are not suitable for RL training of modern LLMs. In contrast, our CGPO introduces **negligible additional overhead** (see the ablations in Section 4.3), making it a more practical and scalable choice for this setting.

---

> ### Author Response · Authors · 2025-11-20
> **Response to Reviewer Fgcd---Part 4/5**
>
> > Q1: How sensitive is CGPO to the number of domains $K$? Does performance plateau or degrade as $K$ becomes very large?
>
> Thank you for your valuable question.
>
> **TL;DR:** CGPO's behavior is governed by the degree of cross-domain conflict rather than by the number of domains $K$. As conflicts grow stronger, CGPO's curvature-informed sequential updates become increasingly beneficial and lead to larger performance gains. Importantly, across all our experiments, we did not observe any evidence that CGPO's performance plateaus or degrades as $K$ increases. Instead, CGPO remains stable even under highly heterogeneous multi-domain settings. We have added new experiments in **Appendix F.2**, and for your convenience we present the main results in the table below (please see the appendix for details).
>
> **Detailed Response:**
> The performance of CGPO is driven primarily by **cross-domain conflict**, not by the absolute number of domains $K$. As discussed in Lines 423-428 of the revised paper, The stronger the cross-domain conflict, the greater the mitigation effect and performance improvement provided by CGPO.
>
> To empirically validate this, we have added three new sets of experiments in **Appendix F.2** that vary in either the number of domains or the degree of cross-domain conflict: (1) math + code (moderate conflict), (2) math + creative Writing (substantial conflict), and (3) a heterogeneous six-domain mixture: math + code + scientific QA + creative writing + logic + tabular. We select FAMO for comparison in these experiments because it is the best-performing baseline at the 7B scale in our main experiments.
>
> For your convenience, the main quantitative results from these experiments are summarized in the tables below.
>
> Table 5: **Performance of models (Qwen2.5-7B-Instruct) trained on the multi-domain dataset (math + code) with different methods, evaluated on multiple benchmarks.** The bold font indicates the best result.
> |          | Math      |           | Code      |           |           |
> | -------- | --------- | --------- | --------- | --------- | --------- |
> |          | MATH500   | AMC       | HumanEval | MBPP      | AVG       |
> | FAMO     | **76.25** | 57.37     | 84.01     | 71.40     | 72.26     |
> | **CGPO** | 76.15     | **60.81** | **84.66** | **72.60** | **73.56** |
>
> Table 6: **Performance of models (Qwen2.5-7B-Instruct) trained on the multi-domain dataset (math + creative writing) with different methods, evaluated on multiple benchmarks.** The bold font indicates the best result.
> |          | Math      |           | Creative Writing |           |
> | -------- | --------- | --------- | ---------------- | --------- |
> |          | MATH500   | AMC       | WritingBench     | AVG       |
> | FAMO     | 74.85     | 54.72     | 64.35            | 64.64     |
> | **CGPO** | **75.10** | **58.94** | **67.01**        | **67.02** |
>
> Table 7: **Performance of models (Qwen2.5-7B-Instruct) trained on the multi-domain dataset (math + code + scientific QA + creative writing + logic + tabular) with different methods, evaluated on multiple benchmarks.** The bold font indicates the best result.
> |          | Math      |           | Code      |           | Science      |           | Creative Writing | Logic     | Tabular   |           |
> | -------- | --------- | --------- | --------- | --------- | ------------ | --------- | ---------------- | --------- | --------- | --------- |
> |          | MATH500   | AMC       | HumanEval | MBPP      | GPQA-diamond | SuperGPQA | WritingBench     | Zebra     | HiTab     | AVG       |
> | FAMO     | **75.30** | 55.02     | 82.93     | 68.60     | 22.64        | 31.58     | 63.09            | 36.84     | 68.71     | 56.08     |
> | **CGPO** | 74.90     | **59.84** | **83.88** | **70.80** | **26.91**    | **31.72** | **65.08**        | **37.63** | **69.57** | **57.81** |
>
> Across all these configurations, CGPO remains **stable and effective**. In particular, the performance gains of CGPO in the math + creative writing setting (2.38) are larger than those in the math + code setting (1.30), confirming our claim that CGPO's advantage grows as cross-domain conflict increases. Moreover, in the six-domain experiment, we also observe a clear and consistent improvement from CGPO, indicating that its benefits persist even as the number of domains increases and the interaction structure becomes more complex.

---

> ### Author Response · Authors · 2025-11-20
> **Response to Reviewer Fgcd---Part 5/5**
>
> > Q2: The method relies on multiple gradient steps per "outer" iteration. While the overhead is shown to be low, are there scenarios (e.g., with extremely large models) where even this could become a bottleneck compared to a single-update method?
>
> Thank you for this thoughtful question.
>
> **TL;DR:** Under standard training regimes for reinforcement learning of LLMs, the multiple gradient steps per "outer" iteration introduced by CGPO are very unlikely to become a practical bottleneck, even for extremely large models. Within each outer iteration, CGPO processes essentially the **same amount of data** as a single-update method; we simply reorganize how this data is used across sequential domain updates. Moreover, the dominant computational cost in RL for LLMs comes from response generation and reward computation, not from the small differences in how many optimizer steps are applied to a fixed batch. We further validate this with timing results on 32B and 72B models.
>
> **Detailed Response:**
>
> (1) **CGPO does not increase the total data processed per outer iteration.** In each outer iteration, CGPO first collects a batch of trajectories from all domains and then performs multiple gradient steps using subsets of this batch. Importantly:
> - The samples in this batch are **used only once** in backpropagation (they are partitioned and scheduled, not repeatedly re-used across inner steps).
> - If we sum over all mini-batches used within the inner loop, the **total number of tokens/examples processed** is essentially the same as in the joint-learning baseline using the same outer batch.
>
> As a result, the overall forward/backward workload per outer iteration is very close to that of a single-update method operating on the same batch.
>
> (2) **In RL for LLMs, rollout cost dominates the training-time budget.** In modern RLHF-style pipelines, the majority of wall-clock time is spent on generating model responses (often thousands of tokens per example) and computing rewards, advantages, and other auxiliary signals. These rollout-related operations typically dominate the runtime, whereas the additional bookkeeping and slightly different scheduling of optimizer steps contribute only a small fraction of the total cost. Consequently, the rearrangement of gradient steps in CGPO does not become a practical bottleneck, because it does not appreciably increase the number of tokens processed or the number of forward passes per sample.
>
> (3) **Extremely large models further amplify rollout dominance.** You specifically ask about "extremely large models". For such models, each generated token is more expensive to compute, and response generation becomes even slower. This amplifies the dominance of rollout cost: the time spent generating trajectories grows significantly, whereas the relative overhead from slightly different gradient scheduling becomes even smaller in percentage terms.
>
> To empirically validate this, we have conducted timing experiments on Qwen2.5-32B-Instruct and Qwen2.5-72B-Instruct, with results shown in the table below. The 32B and 72B experiments are run on rented clusters of 16 and 32 H200 (140GB) GPUs, respectively. Due to our limited compute budget, we report the total and per-step runtime for only the first 50 steps. We sincerely appreciate your understanding of this practical resource constraint.
>
> | Model                | Device       | Methods        | 50 steps (h) | Per step (min) |
> | -------------------- | ------------ | -------------- | ------------ | -------------- |
> | Qwen2.5-32B-Instruct | 16 H200 GPUs | Joint Learning | 7.1          | 8.52           |
> |                      | 16 H200 GPUs | CGPO           | 7.5          | 9.00           |
> | Qwen2.5-72B-Instruct | 32 H200 GPUs | Joint Learning | 12.6         | 15.12          |
> |                      | 32 H200 GPUs | CGPO           | 12.8         | 15.36          |

---

### Official Review · Reviewer_f3Af · 2025-11-04

**Soundness:** 3
**Presentation:** 3
**Contribution:** 2
**Rating:** 6
**Confidence:** 3

**Summary:**

This paper studies multi-domain optimization in RL. The authors propose a simple training design: at each (mini-batch) step, prompts are randomly sampled from all tasks. A random permutation of the task order is generated, and the model performs sequential updates on each task according to this permuted order. They show that the sequential update scheme has a theoretical connection to Newton's method where sequential updates induce cross-domain Hessian–gradient interactions, which encourages cooperation across domains  and steers optimization toward coordinated improvements over all tasks.

**Strengths:**

* Originality: The authors establish a novel connection between task-wise sequential updates within a mini-batch in GRPO and Newton's method and Hessian approximation. This insight and perspective are neat and enlightening.
* Quality/Clarity: The entire paper is very well written, clearly structured, and easy to follow. Overall, it demonstrates high technical and presentation quality.

**Weaknesses:**

* The paper focuses on optimizing the average task accuracy. But in practice, some tasks may be more critical than others, and the proposed method does not have a mechanism to account for task-specific importance or weighting. Extending the framework to handle non-uniform task importance could make the approach more broadly applicable.

**Questions:**

NA

---

> ### Author Response · Authors · 2025-11-20
> **Response to Reviewer f3Af**
>
> Dear Reviewer f3Af,
>
> Thank you for your **valuable review** and for your **high recognition of the originality, quality, and clarity of our work**. Your positive feedback is truly encouraging and means a great deal to us. Below, we provide detailed responses to your insightful and constructive comments. We have incorporated the corresponding discussions into the appendix and added the necessary references in the main text. **All changes are highlighted in blue.**
>
> Throughout the discussion phase, we genuinely look forward to your feedback and are fully committed to addressing any remaining concerns. If our responses have properly resolved your questions, **we would be deeply grateful if you would consider raising your score**. If anything remains unclear or unconvincing, **we would truly appreciate your further guidance**, and we stand ready to engage promptly and constructively.
>
> ---
>
> > W1: The paper focuses on optimizing the average task accuracy. But in practice, some tasks may be more critical than others, and the proposed method does not have a mechanism to account for task-specific importance or weighting. Extending the framework to handle non-uniform task importance could make the approach more broadly applicable.
>
> Thank you for the insightful suggestion.
>
> **TL;DR:** CGPO naturally supports non-uniform task importance by directly incorporating task weights into the optimization objective, without requiring any change to the core algorithm. We have added this discussion to **Appendix E.2** and included a reference to it in Section 2.1.
>
> **Detailed Response:** We agree that in many real-world applications, different tasks may carry different levels of importance. Below we clarify how our framework accommodates such cases.
>
> - **Our current paper focuses on the uniform-weight setting.** In the initial submission, we adopt the simplest objective $\mathcal{J}(\theta) = \frac{1}{K} \sum\_{k=1}^K \mathcal{J}\_k(\theta)$ to highlight the core contribution of CGPO, that is, mitigating cross-domain conflicts and improving multi-domain reasoning performance.
>
> - **CGPO can be directly extended to non-uniform task importance.** If tasks have different priorities in practice, our framework naturally supports a weighted objective. Suppose each task $k$ is assigned an importance weight $w_k$ with $\sum_{k=1}^K w_k = 1$. The objective becomes $\mathcal{J}(\theta;\mathbf{w}) = \sum\_{k=1}^K w\_k \mathcal{J}\_k(\theta) = \frac{1}{K} \sum\_{k=1}^K (K w\_k) \mathcal{J}\_k(\theta)$. This corresponds to multiplying each task-specific loss and gradient by an importance-dependent coefficient. Importantly, **no component of CGPO requires modification**. The algorithm behaves exactly as in the uniform case, but now with user-specified task priorities.
>
> **In summary**, while we focus on the uniform-weight setting for conceptual clarity, CGPO seamlessly extends to settings with non-uniform task importance, making it broadly applicable to practical multi-domain scenarios.

---

### Author Response · Authors · 2025-11-20
**Global Response**

We thank the AC for handling our submission and appreciate all reviewers for their careful review and insightful comments. Their constructive suggestions have greatly improved our work. To help reduce the workload for AC and reviewers, we provide below a concise overview of the feedback and the corresponding revisions.

# Overview of Positive Review Comments
We are honored to have received highly positive feedback from ***all three reviewers***, including:
- **On originality and core insight:**
    - "The authors establish a novel connection between task-wise sequential updates within a mini-batch in GRPO and Newton's method and Hessian approximation. This insight and perspective are neat and enlightening." (Reviewer f3Af)
    - "The core insight of leveraging the geometric structure ... is both innovative and principled." (Reviewer Fgcd)
- **On technical quality, theory, and mathematical foundation:**
    - "The entire paper is very well written, clearly structured, and easy to follow. Overall, it demonstrates high technical and presentation quality." (Reviewer f3Af)
    - "The algorithm is simple in practice and grounded in well-motivated theory." (Reviewer SFav)
    - "The paper provides a solid mathematical foundation for the proposed method. The derivations ... convincingly show how the sequential update scheme approximates cross-domain Hessian-gradient products and, in expectation, encourages gradient alignment." (Reviewer Fgcd)
- **On clarity of writing and presentation:**
    - "The paper is well written; the authors clarify their ideas with a thorough background and intuitive discussion." (Reviewer SFav)
    - "The motivation is clearly laid out, connecting the limitations of existing first-order and gradient-manipulation methods to the potential benefits of second-order information." (Reviewer Fgcd)
- **On methodological contribution and practicality:**
    - "A significant contribution is the design of a method that captures the benefits of curvature information without explicitly computing the Hessian... elegant and computationally lightweight, introducing only negligible overhead ... This makes CGPO highly practical and scalable for large-scale LLM RL training." (Reviewer Fgcd)
- **On experimental design and empirical strength:**
    - "The choice of four distinct domains ... effectively demonstrates the method's ability to handle varied types of reasoning and reward signals... showing that CGPO advances the state-of-the-art." (Reviewer Fgcd)
    - "Evaluating on both 3B and 7B parameter models strengthens the claims ... Using seven established benchmarks provides a robust assessment of multi-domain capability." (Reviewer Fgcd)
- **On ablations and analysis:**
    - "The ablation studies on domain order randomization and the mixing coefficient α are crucial for validating the design choices. The analysis of computational overhead directly addresses a key potential concern for adoption." (Reviewer Fgcd)

# Summary of Reviewer Suggestions and Our Revisions
The reviewer's suggestions and concerns, along with our responses and revisions, are summarized below. We have improved the submission with minimal changes to the main text. The main paper is now extended to 10 pages following the ICLR 2026 guidelines. **All changes are highlighted in blue**.

- **Reviewer f3Af**
  - W1: There should be a discussion on extending CGPO to settings with non-uniform task importance.
    - Added in Appendix E.2 (Pages 22-23).

- **Reviewer Fgcd**
  - W1: The role and intuition behind the final interpolation step should be clarified.
    - Added in Lines 283–293, Section 3.3.
  - W2: The limitations of the paper, including potential bias introduced by using a single LLM-as-a-judge, should be discussed.
    - Added as a standalone paragraph in Section 6. A more detailed explanation is also provided in the rebuttal response.
  - W3: There should be an explanation of why second-order approximation methods are infeasible for RL training of LLMs, and why CGPO is a more appropriate alternative.
    - Added in Appendix E.4 (Pages 24–31).
  - Q1: The sensitivity of CGPO to the number of domains $K$ should be addressed, including whether performance may plateau or degrade as $K$ increases.
    - Added in Appendix F.2 (Pages 32–33), including new experiments.
  - Q2: There should be a discussion of whether CGPO's multiple gradient steps per outer iteration may become a bottleneck in certain scenarios.
    - Added in Appendix F.1 (Page 32), including new timing experiments on 32B and 72B models.
- **Reviewer SFav**
  - W1: The explanation following Eq. (5) regarding the expectation should be elaborated.
    - Revised in Lines 276–281, Section 3.3.
  - W2: There should be a discussion of why joint training cannot induce the same cross-domain effect as CGPO.
    - Added in Appendix E.3 (Pages 23–24).
  - Q1: The potential extension of CGPO to the LLM pre-training stage should be discussed.
    - Added in Appendix E.1 (Pages 22).

---

### Meta-Review · Area_Chair_cWd1 · 2025-12-31

**Summary:**

The authors propose a scalable RL framework that mitigates conflicts between diverse domains (e.g., coding vs. creative writing) by performing randomized sequential updates for each training step. They theoretically prove that this sequential scheme approximates cross-domain Hessian interactions, effectively using the reward surface's curvature to align gradients without the prohibitive cost of calculating the actual Hessian matrix. Empirical results across diverse benchmarks (Math, Coding, Science, Writing) show CGPO achieves faster convergence and higher score than strong baselines like GRPO by encouraging constructive interference between tasks.

Reviewers raised questions about validity of the mathematical formula, missing discussion on limitations, computational bottlenecks and sensitivity to the number of domains, etc. Overall, these concerns have been addressed by the authors, resulting in a solid contribution to the community. Therefore, I will recommend an acceptance.

**Reviewer Concerns:**

Based on the rebuttal and discussion, all reviewer concerns appear to be addressed. There are no remaining outstanding issues.

**Reviewer Scores:**

Reviewer SFav has increase the score from 6 to 8.

Reviewer Fgcd is also likely to increase the score.

---

### Decision · Program_Chairs · 2026-01-26

Accept (Poster)